# High-resolution 4D ERT and below-ground point sensor monitoring of High Arctic deglaciated sediments captures zero curtain effects, freeze-thaw transitions and mid-winter thawing

Mihai O. Cimpoiasu[1], Oliver Kuras[1], Harry Harrison[1], Paul B. Wilkinson[1], Phillip Meldrum[1], Jonathan E. Chambers[1], Dane Liljestrand[2], Carlos Oroza[2], Steven K. Schmidt[3], Pacifica Sommers[3], Lara Vimercati[3], Trevor P. Irons[4], Zhou Lyu[5], Adam Solon[5], and James A. Bradley[5,6]

[1] Environmental and Engineering Geophysics, British Geological Survey, United Kingdom
[2] University of Utah, Salt Lake City, Utah, United States of America
[3] University of Colorado, Boulder, Colorado, United States of America
[4] Montana Technical University, Butte, Montana, United States of America
[5] School of Biological and Behavioural Sciences, Queen Mary University of London, London, UK
[6] Aix Marseille University, Université de Toulon, CNRS, IRD, MIO, Marseille, France

*Correspondence to*: Mihai O. Cimpoiasu (mcim@bgs.ac.uk)

**Abstract.** Arctic regions are under immense pressure from a continuously warming climate. During the winter and shoulder seasons, recently deglaciated sediments are particularly sensitive to human induced warming. Understanding the physical mechanisms and processes that determine soil liquid moisture availability contributes to the way we conceptualize and understand the development and functioning of terrestrial Arctic ecosystems. However, harsh weather and logistical constraints limit opportunities to directly observe subsurface processes year-round, hence automated and uninterrupted strategies of monitoring the coupled heat and water movement in soils are essential. Geoelectrical monitoring using electrical resistivity tomography (ERT) has proven to be an effective method to capture soil moisture distribution in time and space. ERT instrumentation has been adapted for year-round operation in high-latitude weather conditions. We installed two geoelectrical monitoring stations on the forefield of a retreating glacier on Svalbard, consisting of semi-permanent surface ERT arrays and co-located soil sensors, which track seasonal changes in 3D of soil electrical resistivity, moisture and temperature. One of the stations observes recently exposed sediments (5-10 years since deglaciation), whilst the other covers more established sediments (50-60 years since deglaciation). We obtained a one-year continuous measurement record (Oct 2021–Sep 2022), which produced 4D images of soil freeze-thaw transitions with unprecedented detail, allowing us to calculate the velocity of the thawing front in 3D. At its peak, this was found to be 1 m/day for the older sediments and 0.4 m/day for the younger sediments. Records of soil moisture and thermal regime obtained by sensors help define the conditions under which snowmelt takes place. Our data reveal that the freeze-thaw shoulder period, during which the surface soils experienced the zero-curtain effect, lasted 23 days at the site closer to the glacier, but only 6 days for the older sediments. Furthermore, we used unsupervised clustering to classify areas of the soil volume according to their electrical resistivity coefficient of variance, which enables us to understand spatial variations in susceptibility to water phase transition. Novel insights about soil moisture dynamics throughout the spring melt will help parameterize models of biological activity to build a more predictive understanding of newly emerging terrestrial landscapes and their impact on carbon and nutrient cycling.

## 1 Introduction

The Arctic has experienced increases in average air temperature and annual precipitation in recent decades (Rapaić et al., 2015; Rawlins et al., 2010). This has led to a reduction of the permanent ice volume and ice covered area, with the Svalbard Archipelago alone losing approximately 5,000 $km^2$ of terrestrial glacier cover since the beginning of the 20[th] century (Martin-Moreno et al., 2017). Arctic soils are significant regulators of the global carbon budget as they are estimated to store approximately $1035 \pm 150$ Pg C in the uppermost 3 meters – an amount equivalent to roughly a third of the global soil C pool (Hugelius et al., 2014). Glacier retreat exposes sediments which undergo succession (Bradley et al., 2014; Schmidt et al., 2008; Wojcik et al., 2021), with deeper layers of the soil likely to be carbon-limited due to a lack of photosynthetically derived organic carbon (Freeman et al., 2009), and undergoing successional and corresponding physicochemical changes more slowly than surface layers (Rime et al., 2015).

During the winter period Arctic soils are experiencing the highest rates of human-induced climate warming (Graham et al., 2017; Post et al., 2019). Recent studies have observed continuous soil respiration fluxes from Arctic soils during winter periods (Arndt et al., 2020; Natali et al., 2019) as well as bursts of carbon emitted from soils following early spring thawing (Nielsen et al., 2001; Raz-Yaseef et al., 2017; Teepe & Ludwig, 2004), which emphasizes the high vulnerability of these soils to climate change. Records have also captured considerable $CO_2$ emissions from Arctic soils during both of the shoulder seasons – i.e., not only during the springtime thawing, but also during fall freezing (Commane et al., 2017; Zona et al., 2016). Therefore, capturing the timing of the transition periods is important for appropriate parametrization of the net ecosystem carbon balance.

Soil moisture content availability and temperature are known to strongly influence soil biological activity and consequently respiration (Brooks et al., 1997; Elberling & Brandt, 2003; Schmidt et al., 2008). Quantifying moisture variability across pioneer soils along with occurrences of spring snowmelt would greatly improve our understanding of where, when, and for how long, microbial communities are active in these early successional landscapes, and thus can contribute to biologically driven carbon transformations. With time since deglaciation newly emerged sediments are changing their biogeochemical properties (Hodkinson et al., 2003; Kim et al., 2022; Cimpoiasu et al., 2024). Therefore, our understanding of Arctic pedogenesis can greatly be improved by continuous monitoring of moisture availability in sediments at different stages of development. However, repeat surveying or sampling in the Arctic can be challenging due to harsh weather conditions (especially under snow cover during winter months) and distance from population centres; thus, a remote, autonomous method of monitoring such environments is desirable.

The emergence of year-round data (e.g., Boike et al., 2018) will undoubtedly enable an improved understanding and prediction of the fate of Arctic soils under future warming scenarios. Long-term studies have used repeated probing to determine the evolution of active layer thickening (Strand et al., 2021) or microclimate loggers to

quantify spatial variability in near surface temperature (Tyystjärvi et al., 2024). Through a four-year long data series obtained from shallow thermistors (down to 150 cm), Kasprzak and Szymanowski, (2023) verify how topographic parameters determine the spatial variation of near-surface ground temperature on a mountain catchment in southwest Spitsbergen, Svalbard. However, point sensors can only achieve a high spatial resolution over a limited region rather than continuously at the field scale, making it difficult to observe dynamic 3D processes such as water infiltration.

Electrical resistivity tomography (ERT) is a surface geophysical investigation method that provides a fast, cost-effective and minimally invasive way of imaging and monitoring soil moisture in 4D (Michot et al, 2003; Cimpoiasu et al., 2021). ERT has an extensive track record of applications in the field (Loke et al., 2013), including polar environments where it was used for its sensitivity to ice content (Wu et al., 2013, 2017), a weak electrolyte (Farzamian et al., 2020; Hauck et al., 2002; Kasprzak, 2015). Approaches to system development have included both the adaption of commercial resistivity survey instrumentation for monitoring purposes (Daily et al., 2004) and the development of purpose-built resistivity monitoring systems (LaBreque et al., 2004). Hilbich et al., 2011 was one of the first studies that used an automated electrical resistivity instrument to monitor frozen ground. One of such automated ERT monitoring devices, is the PRoactive Infrastructure Monitoring and Evaluation (PRIME) system, successfully used for monitoring hydrogeological processes over space and time, including in remote and cold environments (Uhlemann et al., 2021; Holmes et al., 2022; Cimpoiasu et al., 2024). Alongside ERT arrays, point sensors are often used in order to obtain in-situ calibrations between electrical resistivity and direct measurements of moisture content (Garré et al., 2013).

In this work, in the context of our study site, we address two main questions:

i) Considering the need for year-round measurements of Arctic soil properties, and the vulnerability of Arctic soils during winter and shoulder seasons, (a) can geoelectrical sensor technology be used to continuously monitor the coupled heat and water movement (CHWM) in deglaciated sediments year-round? and (b) can we identify and quantify characteristics of CHWM profile in deglaciated sediments during vulnerable periods?

ii) (a) Considering the need to understand Arctic pedogenesis post deglaciation, can CHWM differences between sediments at different stages of development since deglaciation be identified? and (b) How do they express in relation to physical properties, location and topography, through processes of freeze-thaw transition and melt water infiltration?

We have previously described and characterized the physicochemical properties of two sites in the forefield of Midtre Lovénbreen - a retreating glacier in Svalbard, which we have instrumented with sensor arrays (Cimpoiasu et al., 2024). One of these sites monitors sediments that were exposed by glacial retreat in the last 5-10 years, and the other site monitors sediments deglaciated in the last 50-60 years. Here, we present an in-depth analysis of a year-long (summer 2021– summer 2022) time series of geophysical data obtained from the two sensor arrays, emphasizing the capability to monitor soil water availability across seasons and its implication for sediment development and ultimately pedogenesis. The dataset comprises records of soil temperature, soil water content and snow depth, which complement 4D time-lapse images of subsurface electrical resistivity. We describe patterns in the soil point sensor data regarding an anomalous spring melt event and the spring freeze-thaw transition. We analyzed 4D ERT models in the context of local physical and topographical conditions in order to quantify the location, direction and rate of progress of the spring thaw front and used unsupervised automated clustering in order to classify regions of the subsurface more susceptible to liquid moisture content change, a known driver for soil biological activity.

## 2 Methodology

### 2.1 Study sites

Midtre Lovénbreen (ML) is a polythermal non-surge-type valley glacier (Hambrey et al., 1999) with a north-facing catchment located on the island of Spitsbergen, Svalbard Archipelago (Fig. 1a), approximately 5 km SE of Ny-Ålesund (78°53' N, 11°59' E). The glacier has retreated approximately 1.5 km since its neoglacial maximum attained in 1890 (Hamberg, 1894), uncovering a succession of sediments in different stages of development since deglaciation ("chronosequence"). We selected two sites across the glacier forefield based on the time elapsed since deglaciation, reflecting sediment ages of 5-10 and 50-60 years, respectively. Apart from time, the position on the glacier forefield and topography of the sites are instrumental in shaping the physical properties of the plots (Cimpoiasu et al., 2024). Site 1 sediments are found to be alkaline, with a low cation exchange capacity, large vesicles and minerals trapped in a loam texture. Site 2 sediments, in comparison, contain less silt, but more sand, and have a higher porosity, smaller minerals and vesicles. Site 1 showed a more homogeneous structure with higher moisture availability and saturation whereas Site 2 has revealed a stratified structure with higher moisture variability, potentially explained by greater hydraulic conductivity.

## 2.2 Point sensor measurements

At both locations we augered four ~ 15 cm diameter boreholes of approximately 1 m depth. In each borehole we installed six Teros-11 sensors (24 in total per site). The point sensors were factory calibrated to an accuracy of ±3% over an operational range of -40 to +60 C°. To convert from raw sensor output (*RAW*) to volumetric water content (Ø) the following equation was used:

$$Ø = 3.879 \times 10^{-4} \times RAW \times -0.6956, \tag{1}$$

We did not use any sediment specific calibration curves due to sample unavailability at the time of analyses. The boreholes were backfilled with the sediment extracted during drilling. Overlooking the boreholes, a snow depth sensor was installed. All sensors are attached to a Campbell Scientific CR1000X Datalogger, powered from a 210 Ah battery bank and sustained by a 10 W solar panel. The Teros sensors record hourly data of soil volumetric water content (VWC) and temperature. On each sensor towers (Fig. 1b) we also installed a Browning Recon Force Elite HP4 nature camera, which take daily photographs of the site, from approximately 2 m height, to complement the snow depth readings.

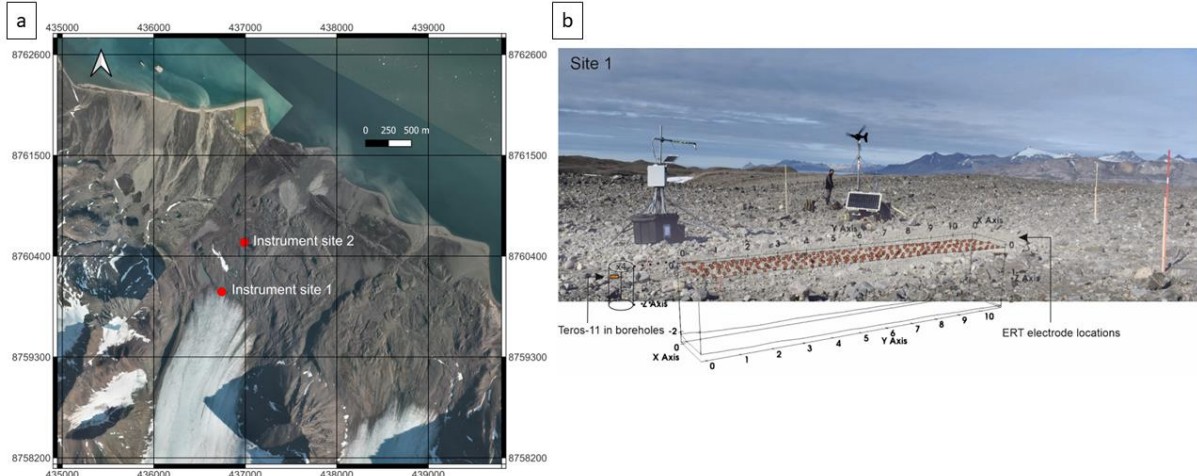

**Figure 1: (a) Topographic map of the study area (© NPI, 2023) (b) Photograph of Site 1 showing the instrumental set-up and the corresponding schematics of the underground sensor installation and the ERT investigation volume (axes units are metres).**

## 2.3 Electrical resistivity tomography

At both sites we installed 222 stainless steel tubular electrodes (high surface area) arranged in 6 lines of 37 electrodes, with 30 cm inline and interline spacing. In the absence of accurate differential GPS positioning, the location and height of the electrodes were recorded manually with a tape measure and laser level. The electrical resistivity measurements are conducted with the PRIME system (Cimpoiasu et al., 2024), which is powered from a 3,780 kWh battery bank. Battery charge is sustained during the summer months by a 150 W solar panel and during the dark winter months by a 300 W wind turbine. One complete set of measurements took approximately four hours, with a measurement frequency of 25 Hz, and contains 12,300 datapoints distributed over a shallow subsurface volume of approximately 11×1.5×2 m³. In order to extend battery life for data collection in the event of a disruption to the power supply, we acquired measurements at a rate of once every four days during the fall and winter months, when the ground is frozen, and changes are minimal. This pattern was followed by more frequent daily measurements starting 1st April 2022 for a greater temporal resolution over the anticipated snow melt period. For each individual measurement, four electrodes are automatically selected by a pre-defined command sequence. As low injection currents and high contact resistances between the electrodes were expected during the winter months (Hilbich et al., 2011; Doetsch et al., 2015), we chose a Multiple Gradient electrode configuration for its spatial resolution properties and greater signal-to-noise ratio (Dahlin and Zhou, 2006). Sensitivity of the measurement sequence was calculated by observing the procedures implemented by Oldenburg and Li, (1999). The full dataset was filtered post-acquisition using the processing workflow detailed in Cimpoiasu et al. (2024). Once pre-processed, the data were inverted using a 4D algorithm implemented in the Res3DInvx64 software from Geotomo (Loke, 2017), as described by Wilkinson et al. (2022).

## 2.4 Relationship between soil electrical resistivity and moisture content

According to Archie's law (Archie, 1942) the electrical resistivity $\rho$ of soils can be expressed as:

$\qquad \rho = \rho_w \, a \, S^{-m} \, \Phi^{-n}$, $\qquad\qquad\qquad\qquad\qquad\qquad\qquad\qquad\qquad\qquad$ (2)

where $\rho_w$ is the resistivity of the pore fluid, $a$ is the tortuosity factor, $S$ is the saturation, $\Phi$ is the porosity and $m$ and $n$ are empirical fitting parameters. Archie's Law is most applicable when the soil and rock matrix can be assumed to be non-electrically conductive. The glacial till sediments in the ML forefield are silty with relatively
low clay content, justifying this assumption (Cimpoiasu et al., 2024).
We divided the ERT volume into three layers of equal thickness, each layer spanning the depth range of two of the six point sensors in every borehole (0-33 cm, 33-66 cm, 66 cm-1m). For every ERT timestep we computed the average electrical resistivity value of each layer and the average corresponding VWC over the day of ERT recording. We then split the range of VWC values into 30 linearly spaced 0.01 size bins (spanning 0 to 0.30 $\text{m}^3/\text{m}^3$)
and computed for every bin the average VWC and its corresponding average electrical resistivity. To these data we fitted Archie's relationship using laboratory-derived porosity values (as listed in Cimpoiasu et al., 2024), an assumed pore fluid resistivity value of 100 $\Omega$m based on previous values of sediment sample EC determined by Kwon et al. (2015) and empirical values of $a$, $m$ and $n$ as fitting parameters.

## 2.5 Movement of the thaw front

Analysing timelapse ERT records, we made the assumption that the onset of soil thaw can be identified by a steep drop in electrical resistivity. This allowed us to select a threshold electrical resistivity value, which marks the point before the drop, of 1,000 $\Omega$m for Site 1 and 5,000 $\Omega$m for Site 2. It is worth noting that because these threshold values were established empirically, if the sediment water saturation were to change a different threshold
electrical resistivity value would be selected. We used these values to track movements of the thawing front in three dimensions, by counting and identifying the location (local Cartesian coordinates, where the origin [0, 0, 0] is one corner of the ERT volume) of all ERT model cells with values below the estimated thresholds. We determined the location of the cell centres by computing an average of the cell coordinates for every timestep, allowing us to calculate distance, rate of progress (gradient of distance) and direction of movement of the thawing
front.

## 2.6 k-means clustering

Unsupervised automated algorithms are useful for identifying patterns in large datasets (Berkhin, 2006). A k-means clustering algorithm partitions $n$ observations into $k$ clusters, in which each observation belongs to the
cluster with the nearest mean (cluster centers) (Berkhin, 2006). This method has been proven to be effective in geoelectrical data classification (Audebert et al., 2014; Giuseppe et al.,2014, 2018). As described by Delforge et al. (2021), time-lapse ERT datasets can be classified after the Coefficient of Variance (*CV*) of every model cell, expressed as:

$\qquad CV = \sigma/\mu$, $\qquad\qquad\qquad\qquad\qquad\qquad\qquad\qquad\qquad\qquad\qquad\qquad$ (3)

where $\sigma$ and $\mu$ are the standard deviation and average, respectively, of electrical resistivity values associated with a model cell. Firstly, an appropriate number of clusters was determined using the elbow method, where inertia (sum of squared distance of samples to their closest cluster center) is plotted against the number of clusters and
the elbow of the obtained curve is selected as the number of clusters to use. Secondly, for every time step, we computed the average resistivity values of all the cells within each cluster and the corresponding standard deviation.





## 3    Results

### 3.1 4D ERT measurements

We have acquired 197 individual timesteps for Site 1 and 200 timesteps for Site 2. Figure 2a shows the distribution of the model cells' sensitivity with values of 0.2 and above. During strong winter storms with high wind speeds,
an overvoltage protection mechanism switched off the PRIME system during the acquisition window on some occasions, resulting in several missing datasets during the winter months. The system was subsequently reconfigured in March 2022, which should minimise this issue in the future.

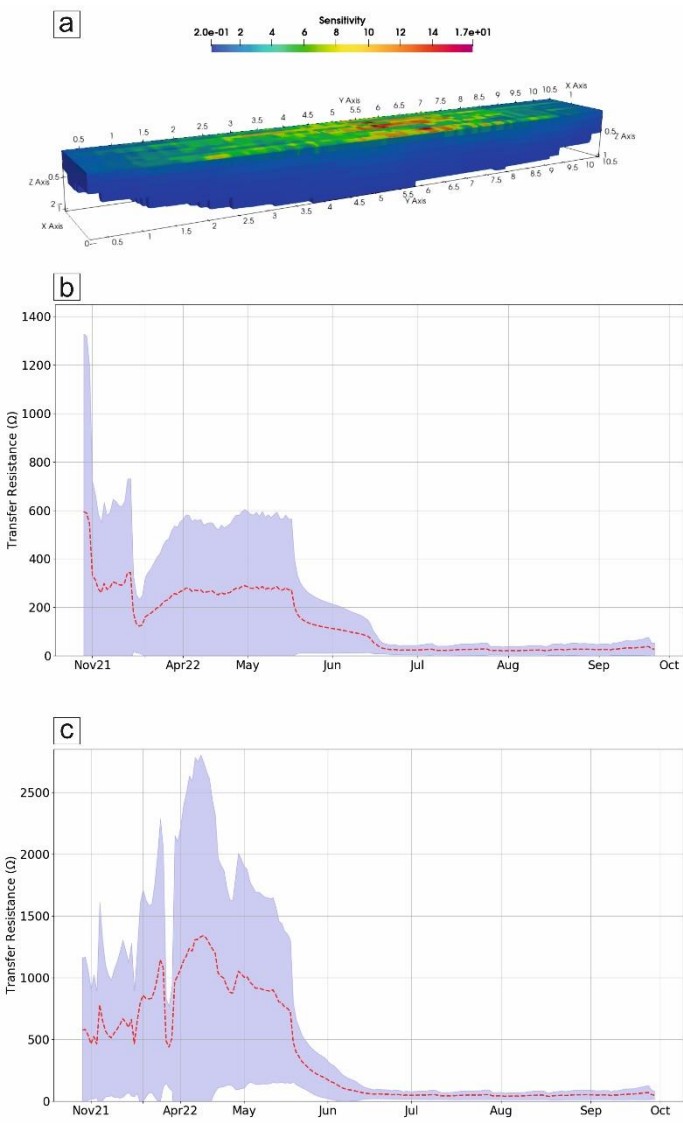

**Figure 2: (a) 3D Sensitivity plot corresponding to the measurement sequence used at both sites. Evolution of transfer**
**resistance (Ω) at (b) Site 1 and (c) Site 2. Red dotted line represents the mean transfer resistance, and the shaded purple area represents the transfer resistance data distribution within one standard deviation.**

During the winter and spring months, we record high values (>500 Ω for Site 1 and >1,000 Ω for Site 2) of transfer resistance (Fig. 2), with Site 2 showing higher overall values. Both sites record a drop in transfer resistance in mid-March followed by a steep recovery. Site 1 records its highest values (1,300 Ω) in late October 2021, whereas Site 2 records its highest values (2,650 Ω) in April 2022. In May 2022, Site 2 experienced a gradual decrease in resistance, whereas Site 1 experienced a two-step drop. Data standard deviation is lower when the average resistance is lower, with Site 2 showing larger overall standard deviations. Contact resistance values ranged between 1-2 kΩ in the summer months and 50-60 kΩ during the winter. Due to high contact resistance values, data retention after filtering and quality control dropped from 97 % during the summer to 62 % during the winter (a complete metadata record is available upon request). Figure 3 shows a selection of inversion results for four ERT timesteps. Data for the first timestep was captured in February when the shallow subsurface is expected to be frozen. The images show elevated values of electrical resistivity at both sites. Site 2 exhibited higher and more uniformly distributed values. A low soil temperature during winter determines strong positive changes in electrical resistivity (in this case the reference dataset was acquired in the summer when the soil is assumed to be unfrozen). The other three timesteps show a progressively thawing soil volume, indicated by the gradual drop in absolute resistivity values throughout the volume and the migration towards zero relative change.

As the sediments thawed, we observe changes in electrical resistivity throughout the respective 3D image volumes recorded at these timepoints (highlighted by purple lines in Figure 3). At Site 2 in the top 40 cm, a distinctive layer of low resistivity forms and extends throughout the volume as time progresses, whereas at Site 1 changes are more uniform. A physically distinct surface layer could be the result of extensive weathering and redeposition of the periglacial deposits, as Site 2 sediments have been exposed approximately 40-45 years longer than the ones found at Site 1.

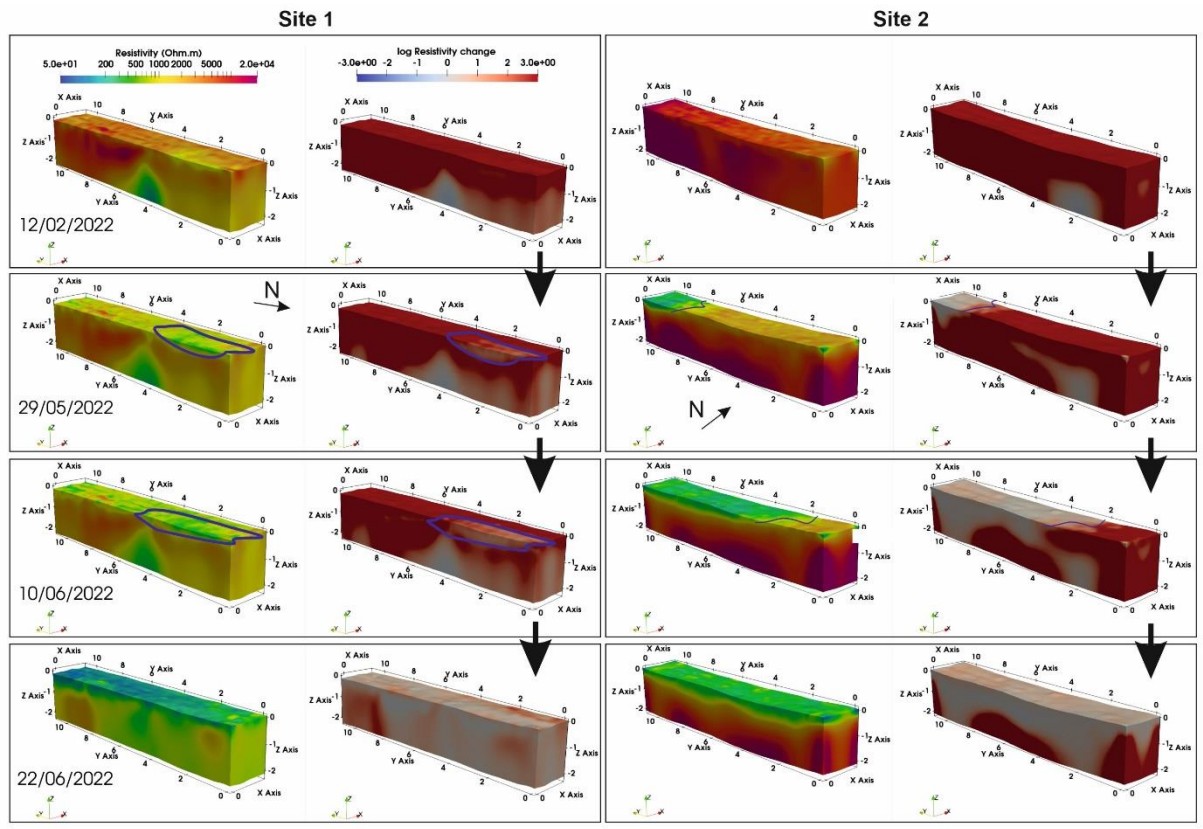

**Figure 3: Inversion results for four selected ERT timesteps. A 3D model of absolute electrical resistivity values is shown on the left-hand side of the subplots and resistivity change relative to ERT data recorded on 2nd Aug 2021 on the right-hand side. Axes are in metres. Purple line delineates zones of interest where changes in resistivity are detected.**

### 3.2 Point sensor measurements

Teros-11 sensors measured continuously from August 2021 to October 2022 (Fig. 4). Both sites experienced data loss due to battery failures. As a result, at Site 1 there are gaps in the data between 3rd Dec 2021 to 7th Mar 2022

and 25th May to 1st June, and at Site 2 between 30th Sep to 26th Oct and between 23rd Dec 2022 to 7th Mar 2022. This meant that the soil freeze, occurring between 10th and 19th of October 2021, could not be captured at Site 2 during this time interval.

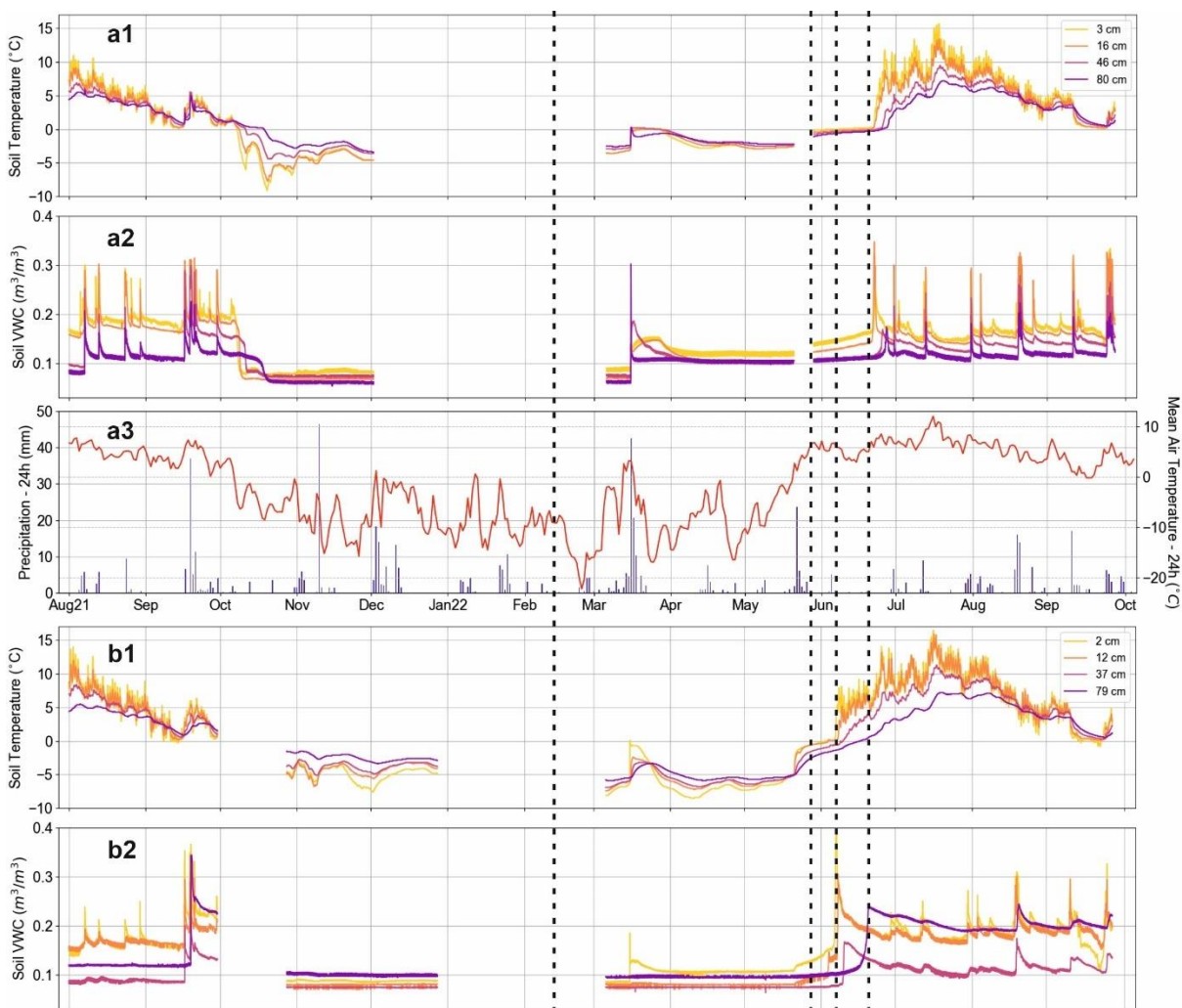

Figure 4: Point sensor data from Sites 1 and 2 showing soil temperature (a1, b1) and soil VWC (a2, b2) between Aug 2021 and Oct 2022, together with air temperature and precipitation (a3) recorded by the Ny-Ålesund weather station (Seklima, 2023). Dotted lines mark the four ERT timesteps shown in Fig. 1.

When the recorded soil temperature rises above 0 °C we observe a steep increase in VWC values as more liquid water becomes available. According to weather records (Fig. 4 a3), air temperature consistently stays above 0 °C starting 17th of May 2022, but this date does not coincide with the soil freeze-thaw transition at either of the sites, even though soil temperature at Site 2 concurrently starts to increase (records unavailable at Site 1). At both sites, thaw onset occurs earlier in the year for the layers closer to the ground surface. Soil temperature decreases with depth at both sites, with Site 2 warmer in the summer months, and Site 1 generally warmer in the winter months. During the summer months, daily variations in temperature are visible in the uppermost two layers and are seemingly attenuated with depth. September 2021 VWC levels are generally similar to summer 2022 levels. VWC spikes during the summer months coincide with large values in the precipitation record. Site 1 sees larger VWC spikes than Site 2 during such events. VWC at Site 2 in the uppermost two layers is generally higher than at Site 1 in 2022.

Both stations have captured an anomalous event in mid-Spring 2022 (Fig. 5), manifested by a sudden increase in soil temperature and VWC, confirmed in the weather records with sustained air temperatures above 0°C for approximately four days and 85 mm of precipitation over the whole week (11th Mar – 18th Mar), with a maximum of 40 mm recorded on the 15th of March. At Site 1 soil temperature has reached 0 °C at all depths monitored,

coinciding with a spike in the VWC recordings. Once the event ended and the temperature dropped below 0 °C, values of VWC were higher than before the event at all depths. At Site 2 only the shallowest sensor detected a soil temperature above 0 °C during the event, which also caused a spike in VWC values at this depth.

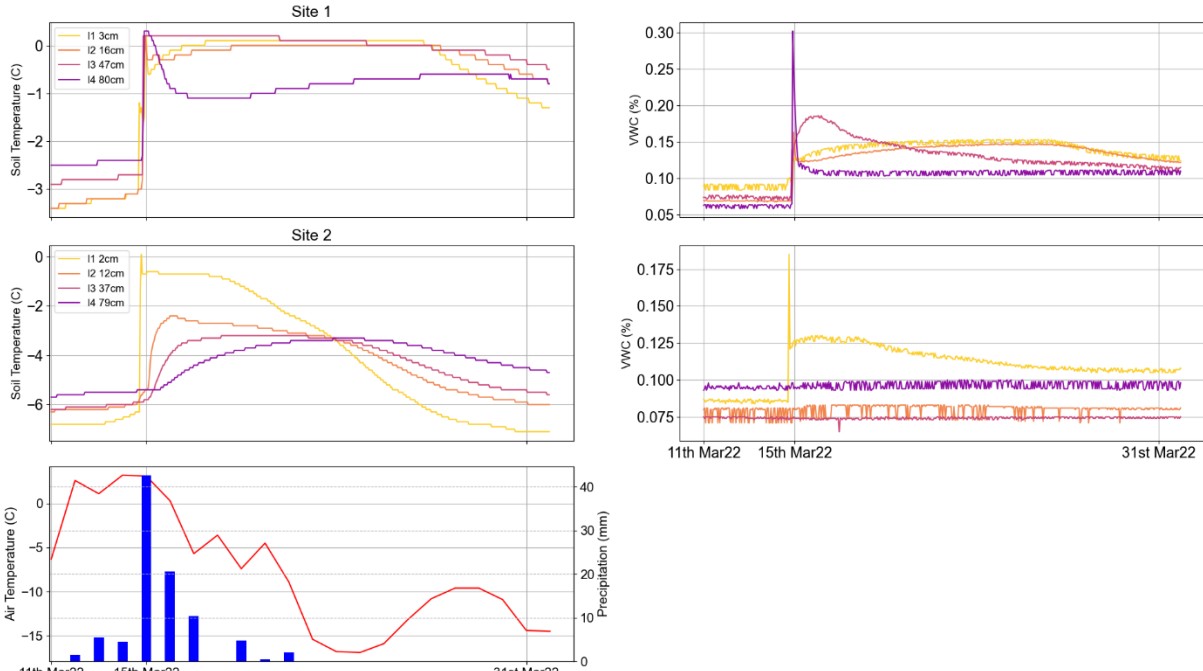

**Figure 5: Soil temperature and soil VWC at Sites 1 and 2 during the mid-March thaw event, shown together with corresponding air temperature (red line) and precipitation (blue bars) records (Seklima, 2023).**

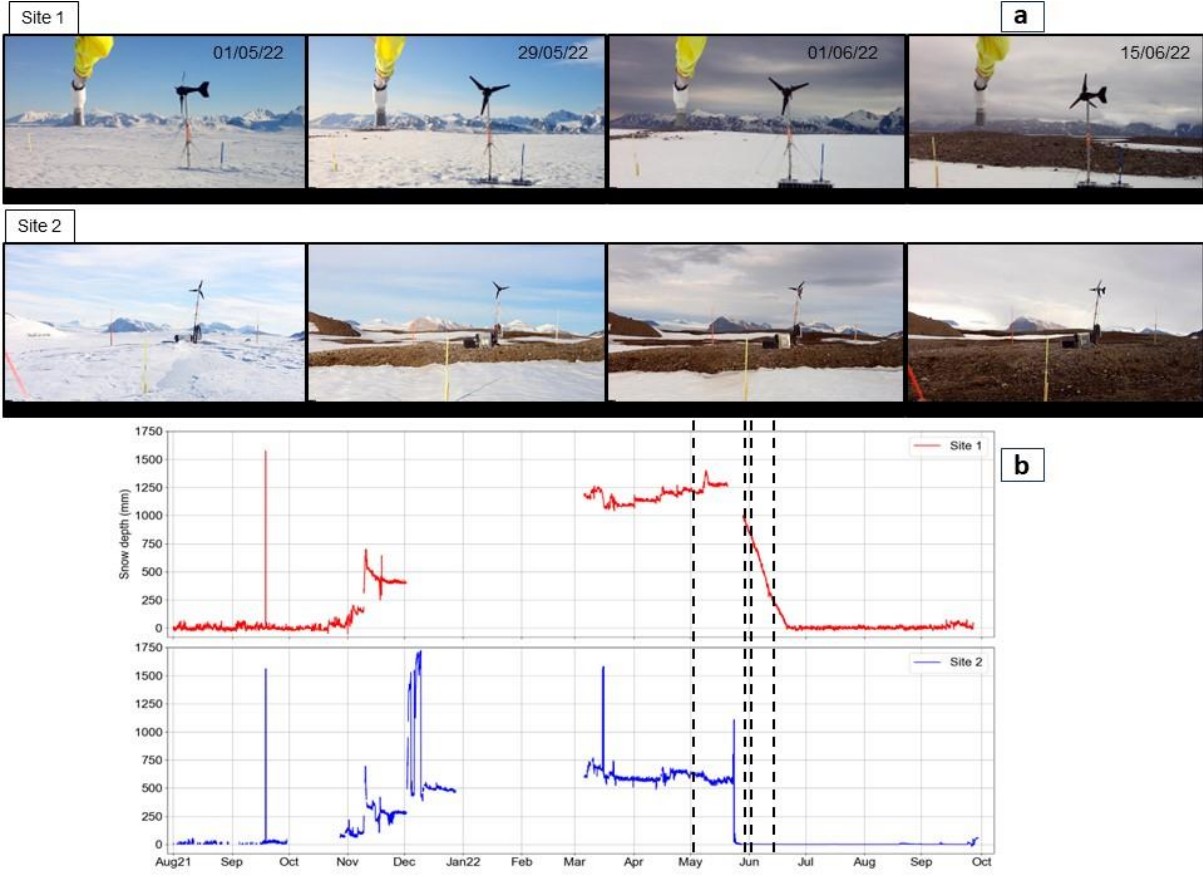

**Figure 6: (a) Nature camera photographs of the two sites. (b) Snow depth measurements corresponding to Sites 1 and 2. Dotted lines in B indicate the dates the photographs were taken.**

We present the freeze-thaw transition above ground in greater detail (Fig. 6), as photographs obtained by the nature cameras display the gradual reduction in snow cover. This visual observation is confirmed above ground by direct measurements of snow depth, and below ground by the buried point sensors (Fig. 7). We recorded a snow depth of 1,250 mm when the snowpack was at its maximum at Site 1 and approximately 600 mm at Site 2. We also observed a faster reduction of snowpack thickness at Site 2 after the air temperatures rose above 0 °C and

a 13-day delay in soil thaw between sites, indicated by steep spikes in soil temperature and VWC values. By contrast, Site 1 shows a more gradual reduction in snow cover. VWC increases due to the freeze-thaw transition occurring on different dates for different soil layers. There is a larger delay between layers at Site 2 as compared to Site 1, with the largest delay of seven days recorded between 37 and 79 cm at Site 2. VWC values spike once the soil temperature rises above 0 °C. At Site 1 we observe greater VWC spikes of 0.15-0.2 m³/m³ recorded by

the top two sensors, compared to the ones recorded by the bottom two sensors of only 0.05 m³/m³. Shortly afterwards (approximately one day), the VWC values revert to values 0.02-0.03 m³/m³ higher than prior to thaw. At Site 2 we observe similar 0.15-0.2 m³/m³ spikes in VWC for the top two sensors, but slightly higher values for

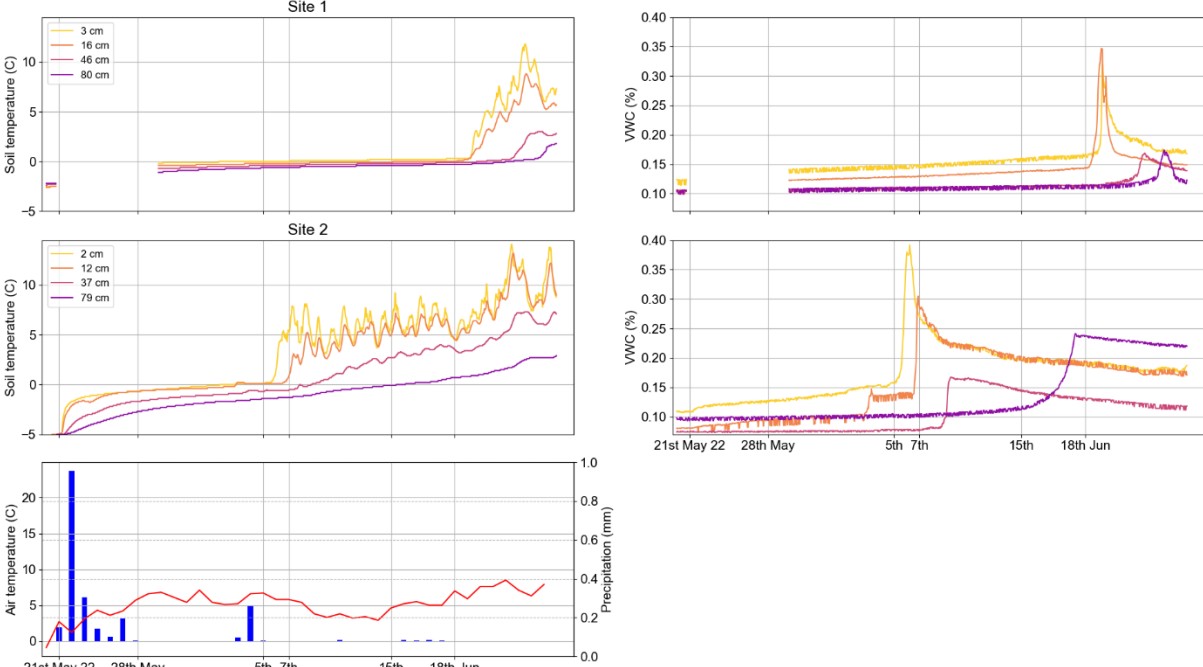

the bottom sensors of 0.07 and 0.18 m³/m³, respectively. After the thaw transition, the decrease in VWC is much slower than at Site 1, taking up to two weeks to record constant values of VWC. Furthermore, the VWC values

post-thaw are higher than the ones recorded before, 0.08 m³/m³ for the top three sensors and 0.12 m³/m³ for the deepest sensor.

**Figure 7: Soil temperature and soil VWC at Sites 1 and 2 during the late May–mid June freeze-thaw transition, together with corresponding air temperature and precipitation records (Seklima, 2023).**

### 3.3 Relationship between electrical resistivity and moisture content

Figure 8 shows the relationship between ER and VWC based on in-situ measurements. Archie's law was fitted with the following parameters: Site 1 layer 1 ($a$=0.36, $m$=2.10, $n$=2.34), Site 1 layer 2 ($a$=0.32, $m$=1.95, $n$=2.58), Site 1 layer 3 ($a$=0.28, $m$=1.85, $n$=2.59), Site 2 layer 1 ($a$=0.55, $m$=2.65, $n$=1), Site 2 layer 2 ($a$=0.58, $m$=2.76 , $n$=1), Site 2 layer 3 ($a$=0.64, $m$=2.94, $n$=1). VWC recordings range between 0.07 and 0.27. Only for Site 2 layer 3 we notice a narrower range of 0.09-0.27. At Site 1 the Archie curves for different layers have similar shapes.

The uppermost layer shows less resistive values when compared to layers 2 and 3, which exhibit a very similar distribution. The calibrated Archie curve misfit is generally low (p-values for layers 1-3: 0.56, 0.62, 0.52), albeit slightly underestimating resistivity values in layers 2 and 3 at high VWCs. Site 2 appears more resistive than Site 1 at equivalent values of VWC. Furthermore, Site 2 shows Archie curves with distinct differences between layers, with values of resistivity increasing with depth throughout the VWC range. The curve does not fit the data as well

as for Site 1 (p-values for layers 1-3: 0.60, 0.46, 0.73), with measurement values in the range 0.14-0.18 not following the Archie model (Laloy et al., 2011).

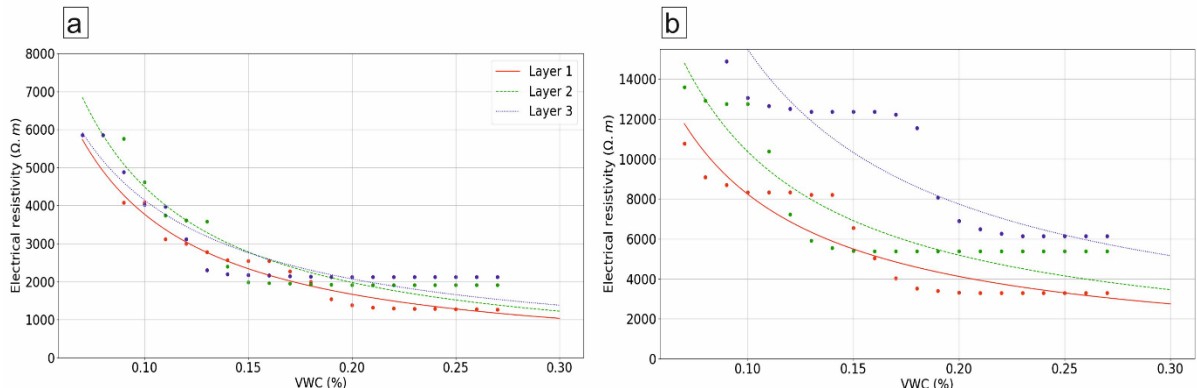

**Figure 8: Best fit of Archie's law (curves) to in-situ measurements of electrical resistivity and soil VWC (dots). (a) Site 1 layers 1-3, (b) Site 2 layers 1-3.**

Archie's law enables us to formulate a site-specific relationship between the electrical resistivity of the observed sediments and their moisture content. We obtained similar relationships with good data fits for the three layers present at Site 1, which is further corroborating evidence for its greater spatial homogeneity. Relationships corresponding to layers 2 and 3 underestimate the resistivity values associated with higher values of moisture content. This is because we had very limited data at that end of the VWC range, with VWC values above 0.2 only recorded during brief summer precipitation events with little impact on the average resistivity of a sediment layer, especially at depth. Archie relationships at Site 2 show a comparatively worse fit due to a data gap during the October 2021 freeze, which could have provided thaw-freeze transitional values of moisture content. Overall, older sediments show greater sensitivity of electrical properties to changes in VWC, which suggests that they dry and freeze faster, shortening the transitional period for microbiological activity within them.

### 3.4 Thawing front progression

Figure 9 presents a visualisation of the thawing front progression at the two sites. The ERT models appear truncated because cells with resistivity values above at chosen thresholds are masked (3.1 at Site 1 and 3.65 log10 Average Electrical Resistivity Ωm). As time progresses, the visible cells have lower resistivity values and new cells emerge, as their resistivities fall below the selected threshold. At Site 2 we observe a clear progression of the thawing front from NE to SW. At Site 1 the progression of the thawing front is not as evident in the timeframe selected as at Site 2, although the emergence of new cells can also be observed. The presence of uneroded boulders and rocks with a different thermal mass than the surrounding glacial diamiction determines an uneven thawing front, especially at Site 1.

The spatial progression of the thaw front is shown in Fig. 10. The localized Cartesian coordinates of the cell centre define the direction of the front. We show the position of the cell centre in the XY plane (a2) and YZ (a3) plane. At Site 1 cell centre moves more in the X direction than at Site 1, whereas the cell centre at Site 2 moves predominantly in the Y direction. At Site 2 the movement of the cell centre starts in late May, marked by a steep movement in the $Y$ direction (along the length of the model) at a rate of up to 1 m/day at its peak (Fig. 10 c2), and in the $Z$ direction (in depth) at a rate of up 0.02 m/day (Fig. 10 c3). Towards the end of the monitoring period, the cell centre tends towards the centre of the model (0.75 m in $X$, 5.5 m in $Y$ and 0.5 m in $Z$) as all the cells have now passed the thaw threshold. For Site 1 we observe two periods of movement; the first concurrent with the movement detected at Site 2, and the second observed in the second half of June. Gradient measurements (Fig. 10 c1-3) indicate that during the first period of displacement the cell centre has moved at a greater rate (up to 0.059 m/day in $X$, 0.41 m/day in $Y$, 0.073 m/day in $Z$) than during the second period (0.018 m/day in $X$, 0.1 m/day in $Z$, 0.04 m/day in $Z$). In the $Z$ direction (Fig. 10 a3, b3 and c3), the cell centre has moved up during the first period of movement and down during the second period. Figure 10 a1 shows thaw cell proportion over time across different layers and the whole model volume. Towards the end of the monitoring window, during the summer when all cells should be below the threshold, the thaw cell proportion is at 100%. At the beginning of the monitoring window, the proportion is not at 0% for all layers shown because some of the model cells had lower resistivities pre-thaw. Site 1, despite seeing a delayed cell proportion increase, reaches 100% proportion before Site 2, which has an initial fast increase phase, followed by a slower gradual increase in the second half of June. Cell proportion dips below 100% in July at Site 1 only to recover towards the end of the month. Surface layers (numbered 1 in Fig.

10) show a faster increase in thaw cell proportion, with Site 2 layer 1 the fastest. By contrast, the deepest layers (numbered 3 in Fig. 10) show a slower increase, with Site 1 layer 3 the slowest.

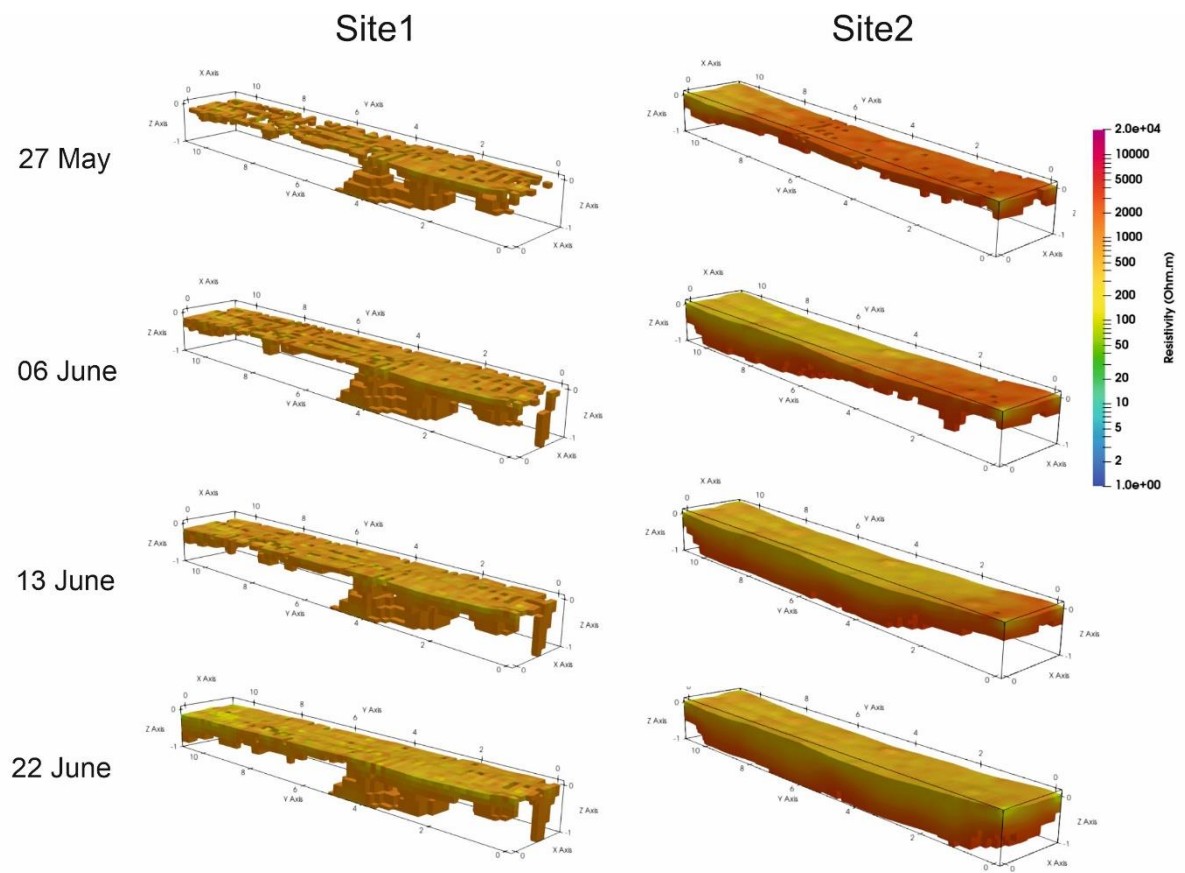

**Figure 9: Four selected ERT timesteps showing only model cells with resistivities below the site-specific thaw threshold values.**

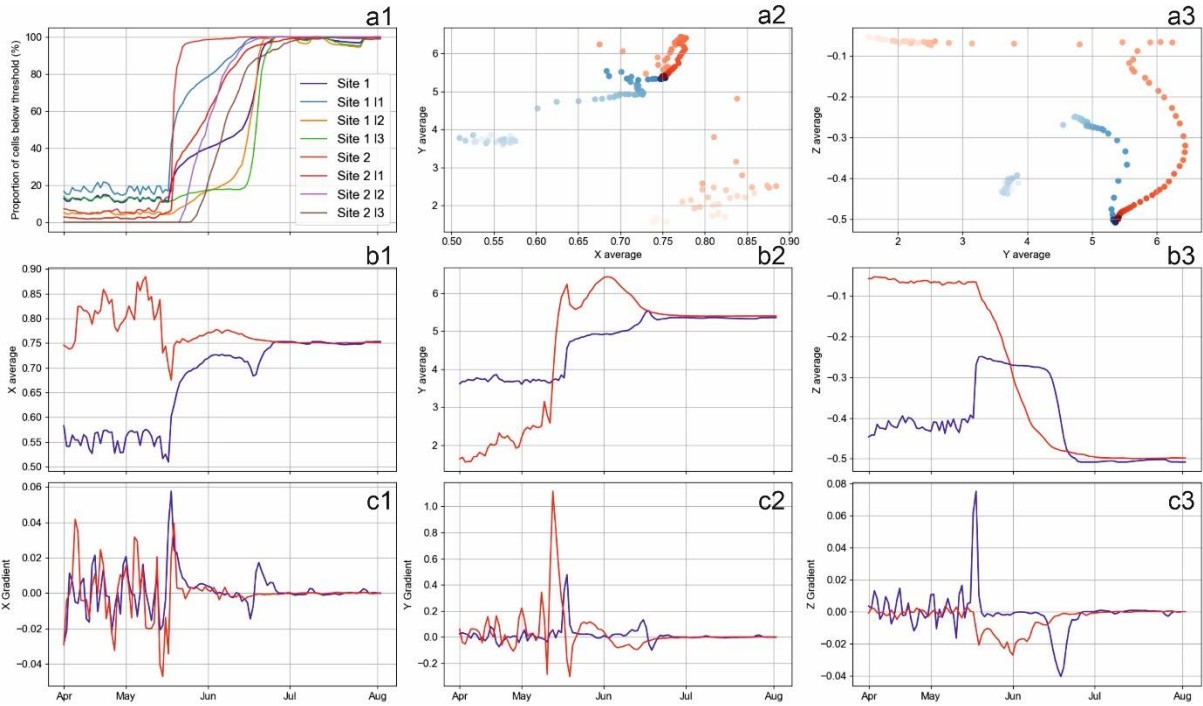

**Figure 10: Thaw cells distribution across the ERT volume. (a1) Number of thaw cells as proportion of the ERT volume and the three layers of equal thickness (l1-l3) described in 2.4. For the following subplots, site 1 is represented by colour blue and site 2 by colour red. (a2) *X* vs *Y* coordinate of the cell centre over time and (a3) *Y* vs *Z* coordinate of the cell centre over time. Fainter colour shades indicate an earlier timestep and stronger colour shades indicate a later timestep. (b1) *X* coordinate, (b2) *Y* coordinate and (b3) *Z* coordinate of the thaw cells' centre. (c1) Movement gradient in *X*, (c2) Movement gradient in *Y* and (c3) Movement gradient in *Z* direction of the model cells' centre.**

### 3.5 k-means clustering

The coefficient of variance (CV*)* distribution within the ERT volume space is shown for both sites in Fig. 11. There are overall higher values of CV at Site 2 than at Site 1, with very high values (>2) at the edge of the model. There are generally higher CV values within the first meter below the model surface at both sites. However, Site 2 shows higher values in some regions below 1 m depth. At Site 1 the median CV value is 0.78 and at Site 2 it is approximately 1. For Site 1, all model cell values are within two standard deviations from the median, whereas for Site 2 there are several model cell values outside this range.

Three clusters have been determined as optimal for the existing datasets (Fig. 12 a3). The spatial division of the model cells by corresponding cluster is shown in Fig. 12 a1, a2. For Site 1, clusters 1 and 3 show similar average resistivity values over time during the winter, whereas cluster 2 shows resistivity values that are lower in comparison. However, post-thaw, during the summer months, all clusters had similar values. Cluster 3 shows the highest values of standard deviation and cluster 2 the lowest. Values of standard deviation are higher during the winter months for all clusters. All clusters also show a similar drop rate in resistivity when soil thaw begins at Site 1 in mid-June. This continues until the last week of June 2022. Furthermore, all clusters show a dip in resistivity during March 2022, coinciding with the anomalous warm event. For Site 2, all clusters show comparatively higher values of average resistivity and standard deviation than the Site 1 clusters during the winter months, with cluster 2 registering the highest values. A drop in average resistivity values in all clusters marks the onset of soil thaw in late May 2022. The drop has a lower rate than at Site 1 and lasts comparatively longer until the first week of July 2022. After thaw onset, cluster 3 shows the highest standard deviation values. Finally, similarly to Site 1, we find a dip in average resistivity values in March 2022.

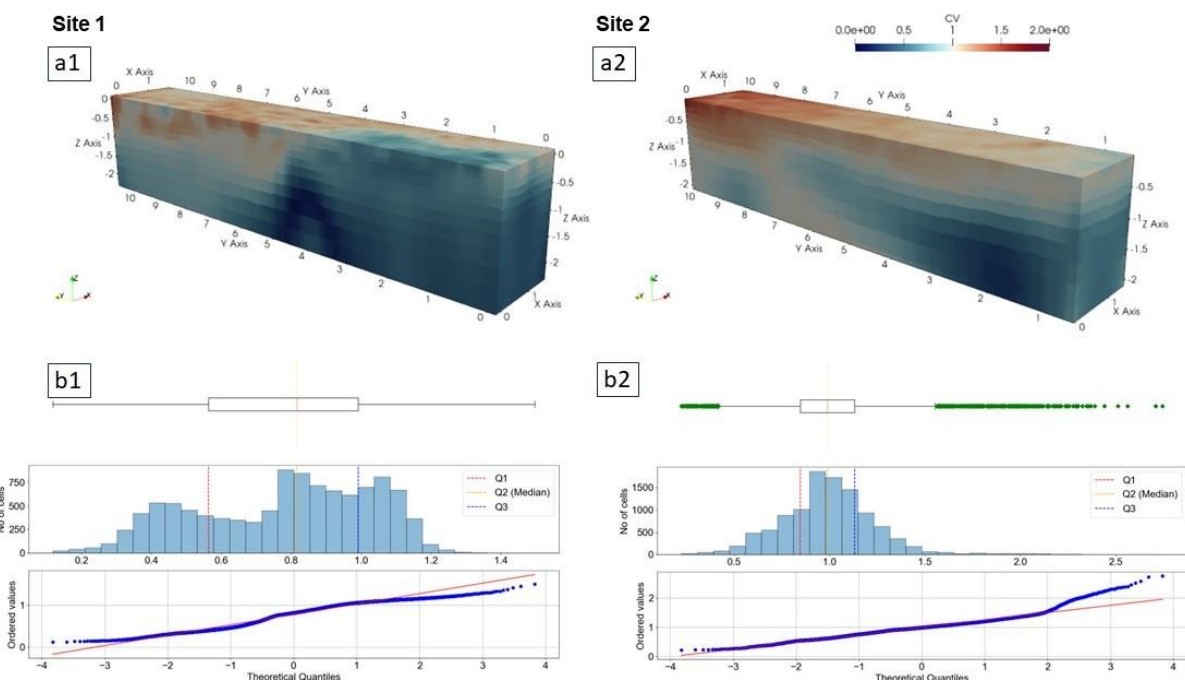

**Figure 11: Spatial distribution of CV values corresponding to (a1) Site 1 and (a2) Site 2. Boxplot, histogram distribution and quantile-quantile plot with respective quartiles of CV values corresponding to (b1) Site 1 and (b2) Site 2.**

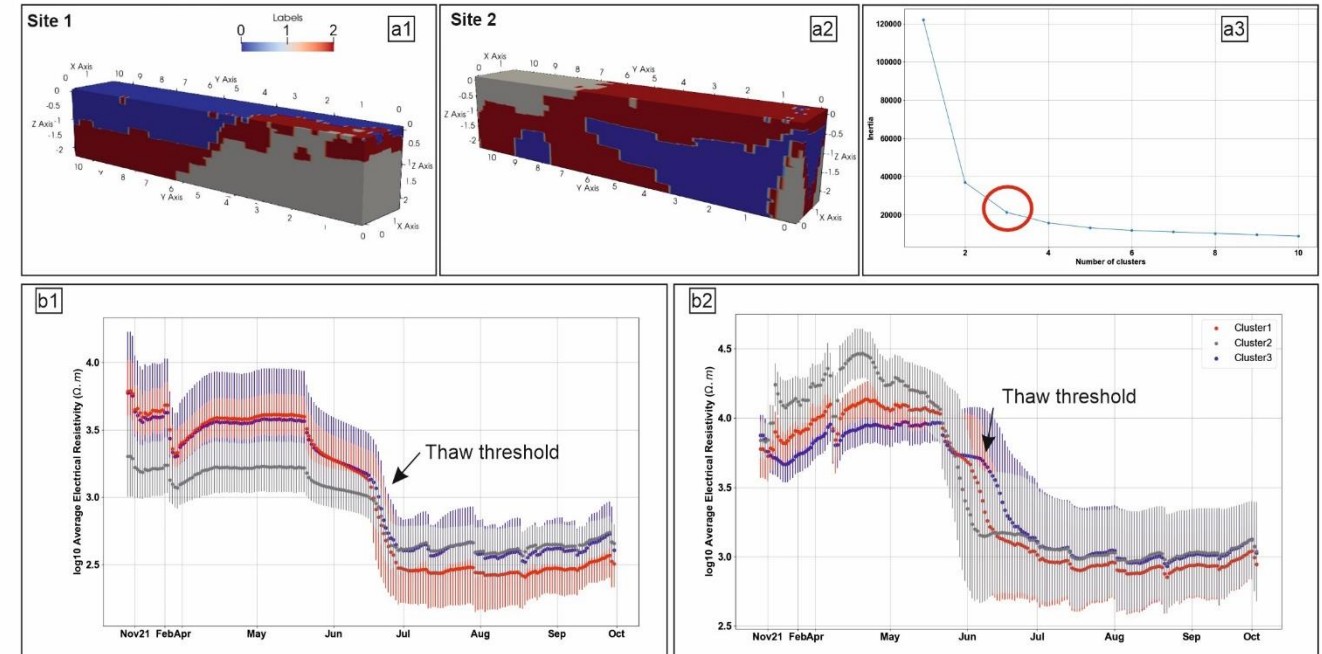

**Figure 12: Spatial cluster division of ERT model cells in reference to their CV value corresponding to (a1) Site 1 and (a2) Site 2. (a3) Results of the elbow method with the chosen number of clusters highlighted by the red circle. Average resistivity and standard deviation values recorded between November 2021 and October 2022 corresponding to (b1) Site 1 and (b2) Site 2 clusters.**

## 4. Discussion

### 4.1 Anomalous snowmelt event

The detection of the snowmelt event in mid-March 2022 presents us with the opportunity to analyze the effects that enabled its occurrence. There were several other instances when air temperature increased above 0 °C (e.g., December 5th, 2021), but there was minimal observed impact on soil temperature or ice content. However, the duration of these events was much shorter than what was recorded in March, when 96 hours of sustained air temperatures above 0 °C caused snowmelt, which occurred during a time of high precipitation. Consequently, melt water infiltrated sediment layers, and we observed an increase in soil temperature and liquid water content. At Site 1, the top 60 cm of the subsurface was maintained at around 0 °C temperature for almost a week, even after air temperatures dropped below freezing. Furthermore, the liquid water content levels of the sediments remained elevated for the rest of the spring. Serreze et al. (2021) recognize the increased occurrence of rain on snow events in the Arctic, where warm spells bring precipitation that percolates through the snowpack and pool at the ground surface. Following the event, air temperature drops below 0 °C, and water refreezes at the ground surface warming the underlying sediments with latent heat, which could explain the elevated levels post the anomalous event of near surface temperature and moisture content.

### 4.2 Soil freeze-thaw transition

Point sensor loggers recorded an extensive dataset over a full yearly cycle, capturing the freeze-to-thaw transition of the sediments present at both study sites. As proposed in Cimpoiasu et al. (2024), proximity to the glacier and local topography is responsible for different thermal gradients and precipitation influxes between sites. This is corroborated by higher soil temperatures at Site 2 and greater VWC spikes at Site 1 during the summer months. Variations in snow accumulation and rate of snowmelt are further evidence of the differences in thermal gradient and precipitation influx between sites. Site 1 experienced greater snow depths and a longer period of snow cover than Site 2, likely driven by generally lower air temperatures at Site 1 that are regulated by the presence of the glacier and the flat topography, which promotes snow accumulation. A prolonged snow cover implies prolonged thermal insulation for underlying sediments, which then led to a delayed freeze-thaw transition at Site 1. This is consistent with the argument put forward by Zhang et al. (2005) that seasonal snow cover can result in an increase of the mean annual ground surface temperature by several degrees and can substantially reduce the seasonal freezing depth. Therefore, younger superficial sediments experience a shorter thaw season, with a delayed freeze-

thaw transition and an early freeze-in the fall due to lower average temperatures. Despite not having intermediary data between the two sites, it is expected that the thaw window increases with distance from the glacier and hence, in this setting, with time since deglaciation.

Water availability is a key driver for biological activity and soil crust development in Arctic regions (Borchhardt et al., 2017). Therefore, the additional 13 days with temperatures below freezing observed at Site 2 may modify local ecosystem dynamics including imposing limitations to soil biological activity and rates of soil development. On the other hand, point sensors detected an increase in soil temperature and VWC under the snowpack even before its melt. This implies that a fraction of the frozen pore water in the sediment transitioned into liquid phase
even before the ground surface was completely exposed. Soil biological activity may occur when liquid water is present, even in pore water – and therefore may persist for a period of 29 days at Site 1 and 15 days at Site 2, respectively, whilst soils remain at temperatures below 0 °C. We therefore consider that soil biogeochemical models applied to the Arctic should consider the variability of water availability across glacial forefields and use such timeseries (if available) in model forcing data and boundary conditions (Lyu et al., 2024).

We observed an intermediate period during the soil freeze-thaw transition when the soil temperature stayed around the 0 °C isotherm (Fig. 7), occurring at Site 1 between 28[th] May–19[th] June (23 days) and at Site 2 between 1[st] June–6[th] June (6 days). This phenomenon is known as the zero-curtain effect and, in this case, occurs when the soil water phase transition to liquid is slowed down by the release of latent heat from the soil to the snowpack (Zhang, 2005). On this basis, a thicker snow layer at Site 1 likely extended the zero-curtain period. Similar zero-
curtain patterns have been observed in the soil temperature data recorded by Boike et al. (2018) at a location a few kilometers north-west from our sites. Here, during the first week of June 2019, the shallow soil (5 cm depth) temperature stayed around the 0 °C isotherm for approximately 4 days.

Monitoring deglaciated sediments over the winter and into the shoulder season completes the CHWM profile of the two sites. Understanding year-round liquid water availability and distribution would not be possible without
coverage of freeze-thaw transitions and the zero-curtain effect.

**4.3 Spatial and temporal behaviour of electrical resistivity**
As shown in Fig. 12, electrical resistivity closely followed the events and patterns observed in the point sensor data. This is to be expected, given that electrical resistivity is strongly influenced by liquid water availability (Samouëlian et al., 2005). Throughout the year, we captured higher electrical resistivity values at Site 2. The
location is warmer and drier during the summer months and colder during the winter months, which would imply less available liquid water for electrical conduction; this is due to a larger proportion of the pore space being air-filled during the summer and ice-filled during the winter. However, liquid water availability is determined by specific substrate physicochemical properties of respective sites detailed in Cimpoiasu et al. (2024) (Site 1 has a lower porosity and a higher silt:sand proportion) which are shown here to impact not only the response to summer
precipitation events but also the way the sediments respond to the freeze-thaw transition (visually represented by the Archie curves).

Four dimensional ERT enables us to track electrical resistivity changes spatially and temporally over the study volume. The resulting models corresponding to both sites contain regions that experience only marginal resistivity changes throughout the year. According to Irvine-Flynn et al. (2011), this could be explained by the presence of
buried ice or large boulders within the matrix of the glacial diamiction and reworked sediments, both of which would be less susceptible to fluctuations in temperature and VWC. In support of this statement, at site, large boulders can often be seen on the ground's surface (Figure 1). Edges and corners of the ERT models show anomalously high values of resistivity, which is especially apparent at Site 2 (Figure 3, 29/05 and 10/06). This can be explained by poorer sensitivity of the sensor array to those regions, with data cover being more limited towards
the outer edges of the model (Figure 2a). Rotem et al. (2023) concludes that recently exposed areas may still go through permafrost aggradation even under the current global warming. However, we could not observe a clear interface boundary between a top active layer and an underlying permafrost. Nonetheless, future geoelectrical installations in similar Arctic settings are encouraged to explore the full extent of the active layer if resolution allows.

**4.4 Tracking the thaw front**
Throughout the winter months, a certain proportion of ice to liquid water content is maintained at a constant level in the pore space throughout the monitored soil volume, as shown by the point measurements of VWC. By attributing a bulk electrical resistivity value to such a constant proportion in the soil volume, we established a

numerical threshold between what can be considered sediments under frozen conditions and sediments undergoing thaw, which will exhibit electrical resistivities below the threshold value. The appearance of new cells in the Fig. 9 timesteps visualises the progression of the thaw front at both sites. Several snow-soil interface cells in the shallow layer (top 20 cm) have electrical resistivity values below the threshold, even before the nominal date when soil temperature rises above freezing. On 6$^{th}$ June at Site 2, we observed that the thaw front had progressed at depth and it continued laterally over the following days. However, at Site 1 we did not observe the same effect; rather the first signs of depth propagation can be seen in the 22$^{nd}$ June timestep, concurrent with point sensor records that mark the first soil temperature above freezing on the 19$^{th}$ June.

Despite commencing later than at Site 2, the thaw at Site 1 covers the entire imaged volume first, as shown in Fig. 10 a1. This can be explained by a greater quantity of snowmelt, which infiltrated into the soil and lowered the bulk electrical resistivity of the shallow subsurface. Furthermore, the flat topography of Site 1 suggests that a greater proportion of the snowmelt will infiltrate the soil rather than generating surface run-off redirected into melt channels. The latter is more likely to occur at Site 2, driven by its hummocky topography.

The Site 1 thaw progression shows two stages of movement; first in mid-May, coinciding with the rise of air temperature above freezing, and then in mid-June when the snow cover disappeared. Between the two stages, the zero-curtain effect maintained the temperature of the shallow subsurface around 0 °C with minimal changes to electrical resistivity. A positive cell centre gradient in $Z$ (Fig. 10 c3) indicates that changes in the first stage of thaw happened at the surface of the monitored volume, followed by a negative change, indicating propagation of thaw vertically downwards. The latter is faster than what we observed at Site 2 (4 cm/day versus 2 cm/day, respectively), illustrating the meltwater infiltration effect discussed above.

Despite being covered by a thinner layer of snow, Site 2 showed generally a faster movement of the thaw front, with the cell center moving by up to 120 cm in one day. However, the movement was primarily lateral (largest gradient values seen in the $Y$-direction – Fig. 10 b2 and c2) along the length of the monitored volume. This highlights the directionality of the snowmelt, which progressed towards the melt channel located at the far end of the electrode array (Fig. 3). Similarly, monitoring a discontinuous permafrost environment in Alaska, Uhlemann et al. (2021) present a record of timelapse 2D ERT images that capture the lateral flow of snowmelt infiltration underlining its role in the formation and development of taliks.

Layers 2 and 3 changed more slowly compared with the surface layer (Fig. 10 a1), and the deepest layer (>70 cm) experienced a reduction in gradient only once all the overlying snow had melted in mid-June. These observations strongly suggest that thaw in the near subsurface is driven primarily by meltwater. Therefore, factors such as topography and snow thickness need to be taken into account when investigating the timing of soil biological activity, particularly at depth. Yi et al. (2015) drew a similar conclusion, as their model results showed that increased snow cover promotes soil respiration at depth.

The ability to track the 4D thaw front, a manifestation of CHWM, underlines that geoelectrical sensor data can be used to monitor seasonal change in deglaciated sediments. Differences in the stage of development (time since deglaciation) may have contributed to the contrasting CHWM behaviour between sites 1 and 2. At an overall warmer site, we observe a more gradual freeze-thaw transition, with less snow cover and a strong lateral flow component of the melt water. Site development is shaped by various physicochemical factors such as erosion and sediment transport or biological factors, microbial activity and vegetation succession. Lane et al. (2017) argues that whilst rapid glacier recession should result in theory in a progressive increase in connectivity of sediment sources to the basin outlet, the supply to capacity ratio does not increase continually with glacier recession. One potential cause is gullying accompanied by the sediment accumulation at the base of moraines that was too coarse to be transported by the proglacial stream, maintaining disconnection of the upper basin, evidence of which we see on the ML moraine complex.

Older sediments have experienced more freeze-thaw cycles since deglaciation. Freeze-thaw processes weaken material for erosion (e.g., Mollaret et al., 2019), however, the presence of water in deglaciated environments is as important as the severity of cold in determining sediment transfer rates (Orwin et al., 2010). Therefore, less liquid moisture availability could counterbalance the effect of repeated freeze-thaw cycles and slowed rates of erosion.

Lastly, previous work suggests that the ML forefield is characterized by a particularly slow rate of soil organic matter accumulation in comparison to surrounding glacier forefields (Wietrzyk-Pelka et al., 2020) and that the rate of soil organic carbon accumulation declines in soils older than 40-60 years (Kim et al., 2022).

## 4.5 CV and unsupervised clustering

Coefficients of Variance at Site 2 were generally greater than at Site 1 (Fig. 11 b2). This can be explained by elevated values of hydraulic conductivity at Site 2. Values outside two standard deviations correspond to cells at the edge of the model, where due to reduced sensitivity electrical resistivity change is overestimated. Regions in both models with very low CV are likely indicating the location of buried boulders or ice, which did not see significant change over the monitoring period. The histogram and quantile-quantile plot in Fig. 11 show a normal distribution at Site 2, whereas those at Site 1 exhibit a more uniform distribution, implying that the younger sediments have a wider range of electrical resistivity variance values. This is yet another metric that could be interpreted in the way that the older sediment site experiences a slowdown in morphological changes and overall sediment evolution. At both sites we notice CV outlier values, towards the high end of the spectrum, manifested by strong deviations from the 1:1 quantile line. The retreating glacier uncovered a mixture of silty and sandy sediments, along with gravels and boulders. Subsurface volumes with varying proportions of sediment to boulders will exhibit different water storage capacities and hence different variance of electrical resistivity. At Site 2 we found more aggregated sediments with smaller and fewer boulders present, resulting from years of erosion. This generates a more uniform spatial change in electrical resistivity (Cimpoiasu et al., 2024).

K-means clustering divided the two monitored volumes into three clusters (Fig. 12 a1-2). The clusters appeared to be localized and did not extend over the entire length or depth of the model. All clusters had large standard deviations during the winter months due to poor data quality, explained by very high contact resistances ($\sim$50 k$\Omega$) between the electrodes and the frozen ground, which allowed only very small currents (0.5 mA) to be injected for each measurement. Site 1 Cluster 3 cells were likely clustered due to the increased moisture content in that region of the model, with the lowest values of electrical resistivity observed during the summer months and the highest during the winter months. Site 1 Cluster 3 it is also the cluster with the largest number of cells, grouping the cells with median values of CV (Fig. 11 b2). Site 1 Cluster 2 represents the region with the smallest change. Site 1 Cluster 1 appears to group cells with intermediate values of CV, however it appears to be a close extension of Cluster 3, given the similarity in average resistivity values. Site 2 Cluster 2 contains cells with the highest values of CV, average resistivity and standard deviation. This far end of the model is in closer proximity to the melt channel and experienced the highest fluctuation during freeze-thaw transitions. The standard deviation in Site 2 Cluster 3 increased right after the spring melt onset, with a peak at the beginning of June, directly correlated with the flow of meltwater perturbing the recordings of electrical resistivity. This cluster has a similar CV and a similar spatial distribution to Cluster 2 of the younger site. However, supporting the observation of different thaw front progressions between sites, Site 2 Cluster 3's decrease in average electrical resistivity comes later in the year with a shallower gradient. This implies that the total variance experienced by the older sediments is similar to what the younger sediments experience but happens over a longer period of time. In a review about freeze-thaw action of rocks, Deprez et al. (2020) list duration as one of the main factors behind freeze-thaw weathering. According to Kurylyk and Watanabe (2020) freezing and thawing processes may not always occur at thermodynamic equilibrium. Disequilibrium pressure can occur during thawing and consequent infiltration, because ice content can decrease without a change in temperature. Therefore, a longer thaw transition window would promote conditions closer to an equilibrium transition.

The clustering algorithm provides an unsupervised method of selecting regions of the model that are more or less dynamic. In our context, it identifies the regions most affected by meltwater infiltration and the regions which are less likely to change, a method of classification with respect to different CHWM profiles. Kasprzak and Szymanowski, (2023) point out that topographic parameters in an Arctic mountain catchment setting determine the spatial variation of near-surface ground temperature. In our study, local variations in topography and glacier proximity determine differences in the way water is routed and retained in the near subsurface. However, this study and Kasprzak and Szymanowski (2023) investigate processes at different scales. We stress that local and regional effects on sediment properties have to be simultaneously evaluated in order to obtain a comprehensive understanding of sediment development.

## Conclusions

A better appraisal of year-round CHWM in recently deglaciated sediments will contribute to a more comprehensive understanding of Arctic soil evolution. In this study we observe two sites in the forefield of the retreating ML glacier, Svalbard, at different stages of development using ERT monitoring technology complemented by point sensors in order to capture deglaciated soil temperature, moisture content and electrical resistivity across one calendar year (Aug 2021–Aug 2022).

Geoelectrical instrumentation was successful in monitoring CHWM processes at both sites. During the Arctic spring, we recorded elevated levels of soil moisture and temperature associated with an anomalous rain on snow event. During the shoulder period between freeze and thaw, deglaciated sediments experienced the zero-curtain effect. The time the sediments spent around the 0 °C isotherm depends on the snow cover thickness, in this instance determined by the local topography and the site's position on the glacier forefield. Finally, our ERT monitoring stations were successful in obtaining almost uninterrupted timelapse recordings, which revealed unprecedented 4D images of the Arctic soil freeze-thaw transition. Such records allow one to calculate the speed, direction, and magnitude of the thawing front. Furthermore, unsupervised k-means clustering proved to be an effective method of classifying regions of the imaged sediment volume according to their electrical resistivity coefficient of variance, indicating how local site conditions affect water storage variability. Some clusters identified are representative of areas of increased water content whereas others are representative of areas containing higher dense materials, potentially buried ice or rocks.

Differences in CHWM profiles between sites are underpinned by site specific Archies law calibrations and different thaw front velocities. We found that the CHWM profile change between sites is expressed through a different thaw propagation, with older sediments dominated by a longer lateral thaw driven by topography and snow, whereas younger sediments were dominated by vertical thaw. Clusters identified at different sites of a similar CV and spatial distribution exhibit a different gradient of average electrical resistivity values, which may again imply that older sediments are subjected to a more time distributed freeze-thaw transition, under conditions closer to thermodynamic equilibrium.

The length of the zero-degree curtain, instances of large melt events, and heterogeneity of soil freezing all have implications for liquid water availability of newly exposed Arctic sediments. As air temperatures and precipitation continue to increase in the High Arctic, leading to further ice loss and exposing more underlying sediments, further research on continuous freeze-thaw dynamics in three-dimensional volumes of soil will be important to better constrain the superficial hydrological regime of the newly exposed landscapes.

**Author contribution**

MOC - Responsible with fieldwork planning, field installation (BGS PRIME), data acquisition, data processing and manuscript writing. OK - Responsible with project management (co-PI), fieldwork planning and manuscript editing. HH - Responsible with fieldwork planning, field installation (BGS PRIME), data acquisition and manuscript editing. PBW - Responsible with initial system design (BGS PRIME), data processing (BGS PRIME) and manuscript editing. PM - Responsible with initial system design (BGS PRIME) and fieldwork planning. JEC – Responsible with fieldwork planning, data interpretation and manuscript editing. DL - Responsible with fieldwork planning, field installation, data acquisition and instrument maintenance (University of Utah point sensors). CO - Responsible with project management (co-PI), initial system design (University of Utah point sensors), fieldwork planning and instrument maintenance. SKS - Responsible with project management (co-PI), fieldwork planning and field installation. PS - Responsible with project management (co-PI), fieldwork planning, field installation and manuscript editing. TPI - Responsible with project management (co-PI), fieldwork planning, field installation and manuscript editing. ZL - Responsible with instrument maintenance. AS – Responsible with instrument maintenance and manuscript editing. JAB - Responsible with project management (co-PI), fieldwork planning, field installation and manuscript editing.

**Data availability**

The Teros-11 sensor data can be downloaded from *Dane Liljestrand, Carlos Oroza, Michael Jarzin Jr, Justin Byington, Zuzana Puc, Trevor Irons, Mihai Cimposiasu, & Harry Harrison. (2023).* *Surface and subsurface hydro-geophysical measurements, Midtre Lovenbreen glacier forefield, Svalbard. Aug 2021 - Oct 2022*. *Arctic Data Center. doi:10.18739/A2PC2TB0B*.

ERT data is available upon request.

**Competing interests**

The authors declare they have no conflict of interests.

**Acknowledgements**

This work was funded by the NSF-UKRI Signals in the Soil program (award numbers, NERC: NE/T010967/1, NE/T010568/1; NSF: 1935651, 2015329, 1935689), and also received financial support from the NERC Covid fund (administered by the British Antarctic Survey). Our work also benefited from Trans-National Access support (project AMBER-ICE) from the European Union's Horizon 2020 project INTERACT, under grant agreement No. 730938 to JB, SIOS (Svalbard Integrated Arctic Earth Observing System) (project: CAP-BIO) to PS, JB, SS, TI, CO, JC and OK, and support from the Agence Nationale de la Recherche (ANR23-CPJ1-0172-01) to JB. We thank the UK NERC Arctic Research Station and the Norwegian Polar Institute Sverdrup Station in Ny-Ålesund, Svalbard, as well as Kings Bay Research for sharing their extensive local knowledge and logistical support. We would also like to thank Juan Carlos Trejos-Espeleta, Justin Byington and Michael Jarzin Jr. for assistance and logistical support in the field. The paper is published with the permission of the Executive Director, British Geological Survey (UKRI-NERC).

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
