# Peer review of "High-resolution 4D ERT and below-ground point sensor monitoring of High Arctic deglaciated sediments captures zero curtain effects, freeze-thaw transitions and mid-winter thawing"

_EGUsphere, 2024_

## Author Response (AR1)

**Reviewer 1.**

**This is a review of "High-resolution 4D ERT monitoring of recently deglaciated sediments undergoing freeze-thaw transitions in the High Arctic" by Cimpoiasu and others. This is an interesting manuscript describing 4D ert measurement at two comparative permafrost sites near a glacier on Svalbard. In general, the writing is clear, though there is room for improvement of organization in some places. The graphics are generally well suited to illustrate the points in the text, though in many cases the text is quite small and a lot of blank space is present between the images. English usage is good. I am very positive about the overall measurement scheme used in this manuscript including 4D ert, many point sensors, met data, and then analysis including statistical clustering – it's really "one stop shopping" for how this type of observation can be made. I look forward to this manuscript being published after appropriate revision so that I may cite it in my own work. Please see general and specific line comments provided below.**

**General comments**

- **Figures have a lot of text/numbers/features that are too small to read clearly.**

R: We have now improved the figures following the reviewer's suggestions.

- **There is no science question or hypothesis stated or tested and therefore it is difficult to evaluate how this work advances knowledge of permafrost hydro(bio)logic systems**

R: We have made changes to our introduction. The scientific aim is to provide a better understanding of moisture availability year-round in young Arctic sediments. Moisture availability is crucial for microbial activity and the development of these sediments. We have also strengthened the way we explain the need for near surface geophysics to achieve such aim and provided a brief overview of other methods of long-term monitoring.

- **It's unclear why so many references to biological activity are made in the intro and discussion, when there do not appear to be any observations of biological parameters made during the experiment.**

R: Biological activity is an essential driver of the evolution and fate of deglaciating Arctic soils. Soil biological activity is constrained by liquid water availability – a property that we can uninterruptedly measure in situ. In our revised manuscript we have included more background context on the role and importance of biological activity, without deviating too much from the focus of our manuscript which is on the geophysical properties of the soil.

- **The discussion section contains quite a bit of test that could be relocated to the results section, and there is a considerable need for better referencing.**

R: Paragraphs indicated have now been relocated to the results section. We have also added references to relevant paragraphs.

- **The conclusion is almost exclusively summary text and does not provide substantial concluding remarks.**

R: We have now redrafted the conclusion in order to capture important remarks.

**Specific comments**

**Line 67: younger/older: does this refer to more recently exposed sediments and sediments exposed longer ago?**

R: Yes. Now rephrased for clarity.

**Figure 1: some of the text on the map is too small to read even when zoomed in on the PDF**

R: Figure has now been amended. The main change was including a different map type.

**Line 115-124: It would be helpful to specifically state the science question and hypothesis in this paragraph.**

R: We have now formulated a clearer aim, namely determining "how much, where and when can we detect liquid moisture (in recently deglaciated sediments) alongside the factors contributing to liquid moisture availability".

**Line 141: How were these Teros11 sensors calibrated for VWC and what temperature range is the calibration valid for?**

R: Details about the calibration are now mentioned in text. Calibration is valid for -40C to +60C.

**Line 160: measurement frequency?**

R: 25 Hz, information now available in text.

**Line 191-193: Please clarify – you assume that the onset of thaw is defined by a "steep" drop in resistivity, which implies a rate of change, however you then go on to use a threshold which does not contain any information about rate. Also, where do the 1000 and 3000 ohmm vales come from?**

R: Threshold values are supposed to mark the point before the drop. This text has been rephrased for clarity: "This allowed us to select a threshold electrical resistivity value, which marks the point before the drop, of 1,000 Ωm for Site 1 and 5,000 Ωm for Site 2. It is worth noting that because these threshold values were established empirically, if the sediment water saturation were to change a different threshold electrical resistivity value would be selected."

**Line 225-235 (including Figure 2): The terminology used here is not standard. I have not encountered the term "apparent resistance" before - is this 1) contact resistance (ohm), 2) transfer resistance (ohm), or 3) apparent resistivity (ohm m)? Also, shouldn't units on contact resistance by kOhm rather than kW?**

R: Now corrected to transfer resistance with the right units.

**Line 312/Figure 9: I suggest to organize this figure similar to Figure 3 with one column for each site and increasing with time downwards. Also, perhaps consider selecting a time period for site 1 where the advance of the thawing front is more evident – the reader does not get much useful information about site 1 spatial/temporal dynamics from this figure. Also, most of the numbers on this figure are too small to read, even when I zoom in on the PDF. There is a lot of unnecessary white space on this figure that could be removed so that the graphics of interest are more readable.**

R: The reviewer suggestions were taken on board and now we believe the figure is more readable. However, we believe the timesteps chosen should not be changed. Firstly, if we were to change the dates on Site1 it would inhibit a direct comparison with Site2. Secondly, Site1 simply does not change in the same way as Site2 does, and we wish this to be reflected in the plots.

**Figure 10: Would it be possible to plot X versus Y with the time encoded as color? Or perhaps X or Y versus Z with the time encoded as color? I like concept of this analysis, however it seems like the way it is presented in this figure is not particularly intuitive for the reader to understand. Even after reviewing the explanation in the Methods section, the concept of "Cell center" in b1 is hard to understand given the plot provided.**

R: We agree with the reviewer, their suggestion will improve the way the reader understands the analysis. We have now plotted X vs Y and Y vs Z and added corresponding explanations in text.

**Figure 12, b1/b2: Resistivity is not a physically linear parameter; it should only be represented in log space. For example, it is non-physical to have a negative resistivity or resistivity of zero. The inset on b1 is too small to read anything.**

R: The figure has now been improved after the reviewer's suggestions. The inset has been removed altogether.

**On the issue of insufficient referencing in various locations.**

R: We have now added further text and references throughout our discussion.

**Line 411: "similar" please provide a quantitative comparison**

R: A quantitative comparison has now been added.

**Line 415: "Fig. 1" Figure number appears to be wrong.**

R: Now corrected.

**Line 417-420: Could this also be attributed to differences in substrate properties? A deeper analysis and comparison of the archies curves and fit parameters may be useful here to support an argument that inherent characteristics of the formation drive differences in average resistivity values rather than just dynamic variables.**

R: We agree with the reviewer and have added this point to the text.

**Line 421-425: Nice job supporting this argument with a literature reference, but is there any evidence measured at the site that would also support this? Did you see any massive ice? What is the typical size of 'boulders' at the site – have you seen ones large enough to explain the regions in the ERT model that do not change?**

R: Yes, boulders can often be found at sites on the surface, with a similar size to what we see. There are photographs of the sites throughout the manuscript that attest this, but the size and location of the boulders, unfortunately, is information that was not quantified. This statement is now included in text.

**Line 425-428: Please calculate the sensitivity of voxels throughout the model domain. For example, Oldenburg and Li 1999 have a pretty general method that should work for this.**

**It would not be necessary to plot the DOI/sensitivity for every pixel on every figure, however it would be useful for the reader to have a quantitative reference that illustrates where the model can be trusted.**

Douglas W. Oldenburg, Yaoguo Li; Estimating depth of investigation in DC resistivity and IP surveys. Geophysics 1999;; 64 (2): 403–416. doi: https://doi.org/10.1190/1.1444545

R: Sensitivity has now been calculated and plotted as a section of Figure 2.

**Line 429-433: This can be moved to the results section. Absent any attempt to draw comparisons with the published literature, this could be considered "interpretation" and may be located with other description of time-lapse results.**

R: Moved to the results section.

**Line 434 – 443: This can be moved to the results section. Absent any attempt to draw comparisons with the published literature, this could be considered "interpretation" and may be located with other description of Archies results.**

R: Moved to the results section.

**Line 511 – 537: This section is currently mostly 'summary' rather than 'conclusions.' I suggest rethinking the content of this section and focusing it on only conclusions.**

R: We have now redrafted the conclusions section.

**Reviewer 2.**

This is a review of "High-resolution 4D ERT monitoring of recently deglaciated sediments undergoing freeze-thaw transitions in the High Arctic" by Cimpoiasu and others. The paper is very well written, and the authors have carried out extensive research trying to decipher freeze-thaw movement. However, I would recommend major revisions. For example, the paper is a case study, but it is written like a report. A report that gives the facts but not too much explanation of why important decisions were made. An issue for example is why specific parameters were utilized. In the Archie's law in 2.4 of the methods, there is no description for the reasoning of the chosen resistivity values. This reasoning needs to be included. I would recommend providing a deeper explanation of both the described values, and a deeper description of all of the figures (more detailed captions). For your case study, the figures need to guide the reader more accurately to what you are trying to convey. I would also perhaps remove some figures such as figure 9 because it is referenced very little and the reasoning for the chosen threshold values is not described well. Furthermore, there are times where it appears that an assumption has been made, as there is no reference to connect to the idea. Certain statements, while understandable, can be made stronger with the inclusion of a reference (e.g., line 402).

**General comments:**

**1The introduction and methods need improvement. There are multiple phases of methodology that have not been previously explained in the introduction. I suggest that the authors add some literature review to the introduction trying to delineate some of their methods and theory behind the study. Concepts such as inversion, clustering (The k-means clustering needs to be better explained for readers who are unfamiliar with the method), etc. must be introduced first.**

R: We have made improvements to our introduction in the attempt to make our aims clearer to the reader. However, we think that notions and strategies regarding data analysis belong to the methods section.

**Line 187-188: Why choose these values for Archie's law?**

R: All the parameters were assigned for specific reasons: Porosity was derived in the laboratory; According to Kwon et al. (2015) that have sampled the forefield at various locations and calculated the sample electrical conductivity, we assumed the pore fluid resistivity to be quite high, 100 $\Omega$m. The other parameters a, n and m are the result of the fit of the Archie curve and our data, they are not chosen per-se. This further justification was added in text.

**Line 193: Why did you choose these threshold values?**

R: We rephrased the text slightly for clarity. These values were chosen to represent the last recorded values of electrical resistivity before a steep drop in our average electrical resistivity data series, which marks the onset of melt.

**2Figures need to be revised throughout the manuscript. Figure captions need to have more explanation and annotation (especially on ERT profiles). Figure 1 should have a better visualization of the ERT electrodes and the ERT frame does not have units. Figure 2 can be placed on 1 singular figure instead of 2. Figure 3 is very small, and the axis do not have a unit. Please separate the study areas to give yourself space to enlarge the figures. Also delineate some zones and give them numbers to discuss in the text. Figure 9 needs a more extensive and elaborative caption. And the color bar does not match the image as some colors are masked on the figure.**

R: Figure 1 has been amended; units specified in caption. For figure 2, we would rather keep the average resistivity distribution in two separate subplots, for better readability. If the editor disagrees, we are happy to change it. Even though we didn't separate the study areas because we think there is remit in showing them side by side, we believe we improved the presentation of Figure 3 significantly, by making the ERT volumes and axis bigger and easier to read. Caption was also amended, and regions of interest highlighted on the figure. Figure 9 and its caption have been amended accordingly.

**Figure 3: Why does site 2 have less resistivity change, but according to the site descriptions the soil has higher hydraulic conductivity?**

R: We are not sure if indeed it has less resistivity change. According to our methodology it has a higher CV overall (Coefficient of Variance). The reviewer suggests a good discussion point which we have included now.

**Figure 5: Explanation of this figure is needed. What is the orange line on the Air temperature figure.**

R: We think the reviewer refers to the $5^{th}$ panel of the figure ($1^{st}$ column $3^{rd}$ row). The blue bars represent precipitation, and the red line represents air temperature. Caption was amended to reflect this more clearly.

**Figure 8: Why did you decide to use these threshold values?**

R: We have responded to this query above.

**Figure 9: Do you need this figure?**

R: Yes, a visual representation of the thawing front is probably one of the most novel parts of this manuscript.

**Figure 10:  The legend. What are "Site 1(1), Site 2 (1)" etc.**

R: We have divided the ERT volume in 3 layers of equal thickness: layer1 (l1) layer 2 (l2) and layer (l3), in order to look at distribution of electrical resistivity values with depth. The caption was amended slightly to reflect this reference more clearly.

**3The discussion section is comprised of many repetitions (from results mostly). I recommend that the authors review this section and attempt to focus more on the reasoning behind the differences between two study sites and try to discuss the causes of variations of ERT values. Another suggestion would be to combine the discussion and result sections and have one result and discussion section to help avoid repetitions. The latter suggestion is at the discretion of the authors.**

R: We have streamlined and relocated parts of the discussion to the results section in order to increase the focus on reasoning.

**Overall, the study is interesting. Congratulations on the good work. Please revise your manuscript in a way that carries more learning potential for the readers rather than reporting your results. Describe the methodologies and interpretation techniques more elaboratively and introduce your terminologies in the introduction and summarize your discussion section.**

R: We thank the reviewer for their encouraging comments. We hope we have revised accordingly.

**Reviewer 3.**

**General comments**

I read the manuscript submitted for review entitled "High-resolution 4D ERT monitoring of recently deglaciated sediments undergoing freeze-thaw transitions in the High Arctic" with attention and interest. The authors used an advanced and, at the same time, sufficiently reliable system for automatic ERT measurements of the ground, enabling repeatable 3D imaging of the two tested sites. Knowing the Arctic realities, this was quite a challenge.

While conducting a literature review for this evaluation, I found an article published in March 2024 by the same authors, "Characterization of a Deglaciated Sediment Chronosequence in the High Arctic Using Near-Surface Geoelectrical Monitoring Methods" (Cimpoiasu et al., 2024). The text contains the research results conducted in the same area, partly in the same sites and dealing with the same problem to some extent. Although it is not an identical material, the submitted manuscript loses some originality. However, I understand the authors' situation. The journal "Parmafrost and Periglacial Processes", in which the article mentioned above was published, restricts the volume of the text and makes it impossible to present all the data collected during the research. So, I guess that the manuscript submitted for review was intended to be an extension of the methodological part of the earlier paper, with more conclusions arising from the ERT method used.

However, I must note that the authors did not formulate the aim of their work. There are two main themes: (i) imaging the thawing and freezing of glacial sediments at two different sites and (ii) examining the thawing front depending on external conditions, including snow cover and the presence of liquid water on the terrain surface. I consider defining the purpose of the study to be crucial in improving the manuscript.

Due to the methods used, observations, and results presented, I consider the topics of gas emissions and biological succession discussed in the text to be unnecessary and disturbing the manuscript structure. However, the introduction lacks a review of the latest research on monitoring the temperature of near-surface ground layers (e.g., Christiansen et al., 2019; Strand et al., 2020; Kasprzak and Szymanowski, 2023; Tyystjärvi et al., 2024; and others) and references to them in the discussion. The description of the methodology also does not convince how original and necessary ERT 3D monitoring is. The choice of research sites is also poorly motivated. The time since the deglaciation of the area is not the only factor that determines the characteristics of glacial till. The differences between the sites result from the sediments' textual and structural features, which the authors write about later in the work.

**The authors do not mention the occurrence of permafrost in the marginal zone of Midtre Lovénbreen (Glacier) at Kongsfjord. Considering the relatively quick formation of permafrost (Rotem et al., 2023), it can be assumed that it may occur at least in one of the studied sites. From my point of view, the planned installation should be used to image the entire thickness of the active layer (whether on permafrost or in meaning seasonally frozen ground), although probably at the expense of model resolution. Please discuss this matter.**

**I do not doubt that the research is a further step in understanding the characteristics of seasonally frozen ground. I appreciate the effort put into obtaining the data and processing it.**

R: The article submitted to TC has a different aim and scope to the one published in PPP earlier this year. The PPP paper is a characterization of the sites and the thermal and hydrological changes over a short summer interval (2 months). The TC paper provides a more in-depth analysis of a longer-term record of ERT data (an entire year vs 2 months). Specifically, a continuous year-long dataset allows us to focuses on the capability to monitor soil water availability across seasons, with a special focus to the spring freeze-thaw transition – which is much more lacking in the literature and for which we have much less data and knowledge.

Following the reviewer's suggestion we have now supplemented the introduction with a paragraph about monitoring the temperature of near surface layers using the references suggested. Also, we have strengthened the necessity and novelty of 3D ERT by emphasizing its advantages over the use of point sensors.

We have now included a paragraph in the discussion which refers to permafrost. However, imaging the active layer did not fall within the scope of this study. As is, the electrodes are not placed sufficiently far apart to allow the PRIME system to image the interface between the active layer and permafrost.

**Specific comments**

**Page 2, lines 81–86: I doubt whether the landscape can be covered with glacial till; it is probably about the terrain's surface. There are too many inaccuracies and generalisations in all these sentences that are difficult to verify. "…Deeper ground layers undergo successional changes more slowly than surface layers… " - What changes? The soil-forming process? This is probably too trivial to write about.**

R: The sentence regarding glacial till has been removed. Succession is the process by which natural communities replace (or success) one another over time.  The other sentence is rephrased to "with deeper layers of the soil likely to be carbon-limited due to a lack of photosynthetically derived organic carbon (Freeman et al., 2009), and undergoing successional and corresponding physicochemical changes more slowly than surface layers".

**Page 2, lines 88–94: I am afraid this information does not apply to glacial sediments examined in the presented manuscript but to thicker sediments and mainly soils covered**

with tundra or even peat bogs. The information provided also applies to permafrost, which, as I understand it, was not found in the studied sites (?). There is some inconsistency here.

R: The sentence referring to thawing of permafrost and destabilisation has been removed to avoid the inconsistency reported.

**Page 2, lines 109–111: It would be good to tell the reader when the first continuous ERT monitoring systems appeared (used by whom). In the literature review, it is good to notice some groundbreaking research, e.g. the first applications of ERT in the study of permafrost or glacial sediments, the first measurements with high resolution (small electrode spacing), etc.**

R: Following the reviewers' suggestion, relevant references were added to the introduction.

**Page 3, lines 133–135: For studies designed in this way, information on the sediments at both sites should be more detailed and supported by laboratory tests. However, this is a non-binding note. More information is provided in the paper published in the PPP journal.**

R: As the reviewer pointed out, we have published physicochemical information about the sediments in a different paper which we referenced to in this manuscript.

**Page 5, lines 226: How exactly were the resistivity limit values established for the ground subject to thawing/freezing? Was it a chart analysis (graphical method)? What I miss in the further discussion part is the consideration of how these values may change for the same site with different water saturation (?), which is theoretically possible.**

R: We rephrased the text slightly for clarity. These values were chosen to represent the last recorded values of electrical resistivity before a steep drop in our average electrical resistivity data series, which marks the onset of melt. This value would change from season to season if the water saturation were different. We have added this point to the discussion.

**Page 6, lines 237–238: The study does not show the results of the sediment test. Therefore, it cannot be assumed that ground ice was formed in the ground profile; maybe it was only the effect of lowering the temperature (?).**

R: We agree with the reviewer the sentence was speculative, it has now been rephrased: "A low soil temperature during winter determines strong positive changes in electrical resistivity".

**Page 7, line 255: "Soil thawing occurs at each site when soil temperature rises above 0°C". Please change this sentence because it represents a truism.**

R: Sentence was rephrased. "When the recorded soil temperature rises above 0 °C we observe a steep increase in VWC values as more liquid water becomes available".

**Page 11, Figure 9: This is the most interesting figure in the manuscript. Has it been stated which ground factor is responsible for the uneven thawing front, especially at site 1? Please rearrange the figure so that the block diagrams for each site are next to each other, e.g., site 1 on the left and site 2 on the right.**

R: Figure 9 was now modified. We believe that the uneven thawing front at Site 1 might be due to the presence of boulders and uneroded rocks with a different thermal mass than the surrounding glacial diamiction. Information added in text.

**Page 13, lines 376–377: Please correct this inaccuracy in the text. I guess it's about analysing the effects, not the conditions (?).**

R: Sentence now amended.

**Page 14, lines 409–411: "According to Elberling and Brandt (2003), microbial activity may continue in unfrozen soils whilst the diffusion and advection…" – I do not understand this sudden change of topic, and I consider this sentence unnecessary.**

R: We agree, the sentence was removed.

**Page 16, lines 530–537: I suggest that this fragment of the conclusion be either deleted entirely or rewritten. In its current form, it contains oversimplifications and numerous logical errors. It also directly refers to issues that have not been the subject of research (soil biological activity, carbon cycle in the environment).**

R: The conclusions section was restructured and rewritten. Portions of the paragraph indicated were either removed or rephrased in order to reduce references to soil biological activity.

**Technical corrections**

**The manuscript must be checked for technical correctness (citations, etc.). I do not see any significant errors, but they may be pointed out during editorial work.**

R: We are pleased to see that the reviewer does not see any major technical errors. We shall amend them if any become apparent during a potential future editorial review.

---

## Author Response (AR2)

**C:** In many places the revisions have substantially improved the work. The reorganization of the text and enhancements to the figures are most welcome. In other areas, the revisions fell short and were somewhat cursory. There is still not really a science question or hypothesis. Key conclusion content is left unsupported by content in the body of the manuscript. Interpretation of data in terms of physical process explanations and comparisons between the study plots is limited. As reviewers commented during the first round, the manuscript still reads much a like a report rather than a scientific manuscript explaining how a cryosphere process works.

The science question remains weak. It appears to be: "We therefore seek to unravel how much, where and when liquid water can be detected alongside the factors contributing to liquid moisture availability." Therefore, the question appears to be 'How much water can be detected?' This does not seem to lead to the compelling investigation of an environmental process. Furthermore, a hypothesis still is not present. The preceding sentence in the introduction describing the aim of the work appears ripe for designing an interesting comparative experiment, however a hypothesis exploring the cryosphere processes within the scope of measurements is not proposed.

**R:** We thank the reviewer for acknowledging the improvements we have done. Our work has been improved so that it does not read like a report, as was acknowledged by the other reviewers who had no further comments. The comparisons between the study plots are everywhere throughout the manuscript. We are not sure why the reviewer sees them as so weak in substance whereas we see them as quite conclusive in terms of what they are trying to show. In the light of this we are trying to push forward the following hypothesis (line 127): "we aim to test whether, in the context of our study site, the hydrothermal profile of sediments changes in relation to physical properties, location and topography, and if, with increased time since deglaciation, sediments exhibit a more gradual freeze-thaw transition and a slower melt water infiltration." We have added and edited portions of the text (e.g., Line 508-512, 603-618, 718-727) to reflect this and supported it with relevant explanations of the underlying processes. We believe that sediments deglaciated for a longer period of time have a higher thaw window during the summer season.

Finally, if the reviewer could be more specific about what conclusion content is unsupported, we would be happy to adjust it accordingly. Nonetheless, we have made adjustments to our conclusion section in order to better reflect our results and interpretation.

**C:** Line 325 – 350 & figure 8: This is a nice analysis, however it is stated earlier in the methods that the VWC probes are not actually calibrated to in-situ soils but rather use the factory calibration (Line 153). This being the case, what value is there to 'calibrate' the ERT data to the VWC data that are not tied to any reality at the site?

**R:** We understand the reviewer's concerns. However, we consider that the factory calibration used is generic enough to cover the type of sediment observed on our sites in Svalbard. We have now mentioned this as a potential source of error.

In figure 8 and related text we associate PRIME electrical resistivity measurements acquired independent of the point sensor measurements of volumetric water content. We believe it is

indeed a site-specific pedophysical relationship because properties of the sediments reflect in the shape of the Archie curves.

**C:** Line 447-453: The purpose of this text is not clear. What argument is it attempting to support? What explanation for the observations is provided? Referencing is inadequate.

**R**: The purpose of this paragraph was to discuss the importance of our findings, i.e. there is a difference of 13 days between the melt onset at the two different sites. We think there is a clear indication of variability in water availability across the forefield which consequently impacts the rate of microbial activity and ultimately soil evolution. These findings/timeseries can feed into future biogeochemical models. We have now rephrased the paragraph to make its message clearer.

**C:** Line 484: Is the boundary between active layer and permafrost at these sites known from other measurements? Could this be compared to the ERT data? Why would the 'vertical resolution" prevent observation of this transition? CALM data around Svalbard seem to indicate that ALT on the island can range from just over half a meter to in excess of 2m, so it is plausible that the entire ERT volume is in the active layer, if permafrost is present at all.

**R**: We understand how the sentence might be confusing. Yes, the interface between the active and the permafrost layer is known from other sources, which we considered beyond the scope of this work. We have now reformulated the sentence in order to remove ambiguity.

**C:** The analyses of the k-means clustering remains limited. While it is claimed that the three identified zones represent similar dynamics within each zone, the timeseries (Figure 12 b1 b2) cast doubt on their being any significant differences between the zones within each site. It is claimed on Line 551 that "The clustering algorithm provides an unsupervised method of selecting regions of the model that are more or less dynamic." However, there is no evidence of this in the timeseries (dynamic) data in figure 12. All three clusters at each site appear to have roughly the same scale of dynamics, though small variations in timing may be apparent.

**R**: Timeseries in figure 12 show the evolution of average resistivity values per cluster over time. We believe the timeseries do not overlap and are distinct enough, therefore showcasing differences between clusters. In our results and discussion sections we tried to point towards these differences in text (e.g., Line 461-463: "For Site 1, clusters 1 and 3 show similar average resistivity values over time during the winter, whereas cluster 2 shows resistivity values that are lower in comparison", Line 712: "Site 1 Cluster 2 represents the region with the smallest change" etc.). Nevertheless, we have added new discussion text to provide more substance to our analysis (Line 718-727), focusing on how sediments at the two sites observed experience thaw transitions of different lengths and how this is reflected into the electrical resistivity variance data.

**C:** Line 539: contact resistance units should be in ohms rather than kW.

**R**: Now amended in text.

**C:** Line 565 – 566: This is misleading because as stated in the manuscript manufacturer equations were used for the VWC data, so no site-specific calibration has actually been made.

**R**: We responded to this comment above.

**C:** Line 575 -576: This statement about the "older" versus "younger" sediments is not argued in the manuscript text. It only appears in the abstract and the conclusions and no evidence is provided to support it. Why would older or younger sediments have this physical effect on water dynamics in the subsurface?

**R**: We now added a few more references to the age of the sediments in our latest attempt to supplement the discussion with a paragraph about sediment evolution rate.

Editor's comment on Reviewer 3

**C:** Reviewer 3 raised no further objections to publishing the manuscript but noted concerns about overlaps with a previous publication in "Permafrost and Periglacial Processes." To address this, I suggest incorporating your earlier response to this comment into the introduction of the revised version and clarifying the aim and findings of the previous publication.

**R**: The response to that comment was added to the introduction and the difference in aim clearly stated.

Other comments:

**C:** Please check your figures using the Coblis – Color Blindness Simulator (https://www.color-blindness.com/coblis-color-blindness-simulator/) and revise the colour schemes accordingly.=>Fig. 8

**R**: Line styles in Fig. 8 were changed to accommodate color blindness.

---

## Author Response (AR3)

Editor's comments:

C: After reviewing the latest round of feedback from referee #1, and examining the revisions made in this version, I share the concerns raised regarding the manuscript's scientific structure and the absence of a clearly articulated science question. Referee #1 has noted that the lack of a science question has led to a disconnection between the outcomes presented in the manuscript and the conclusions drawn which was not well addressed in the revised version. At this stage, restructuring the manuscript around a central science question, as emphasized in the reviewer's comments, would significantly improve the clarity, direction, and overall scientific contribution of your work.

R: We thank the editor for their continuous support. We appreciate the concerns raised by the reviewer and acknowledge the importance of a clearly articulated science question. In this latest version of the manuscript, we restructured the narrative of the introduction which now leads to two science questions (please refer to the reviewer's comments for a detailed description of the narrative and associated science questions). Furthermore, we reformulated the manuscript's conclusions in order to reflect answers to the central science questions enabled by the outcomes of our work (please refer to the reviewer's comments for a full list of outcomes and their corresponding reference in the conclusions text).

Reviewer's comments:

C1: A clearly articulated science question is the only way that readers can understand: 1) what question are the authors trying to answer? And 2) after the experiment, did the authors indeed answer the question? The science question is also essential to judge if the hypotheses are plausible, and often helps to support judgement if the methods and experimental design are appropriate. Importantly, the science question also enables judgement of the significance of the experiment/results. Because it is not defined what question is being asked, perhaps it is not surprising that the results/outcomes of the work do not clearly lead directly to conclusions. If it is not clear what question is being asked, how could we possibly know if it has been answered?

R1: We understand the reviewer's concerns and we have now reformulated the text in the introduction section.

We would like to summarise the structure of the introduction here (line numbers refer to the trackchanges document), highlighting in red areas where we think there is a gap in knowledge and in blue our proposed solution.

L79-L88: Deglaciation and the emergence of soils in the Arctic.

L89-L97: During winter, Arctic soils are experiencing the highest human induced climate warming. Evidence of soil respiration detected during winter and shoulder seasons emphasises Arctic soil vulnerability to a changing climate.

L98-L108: Soil moisture a driver for biological activity. As fieldwork is restricted to the summer period, there is a current gap in soil moisture data due to weather conditions. Also, as soil moisture availability is key for understanding pedogenesis, appraisal of sediments at different stages of development is required.

L109-L121: Emergence of year-round point sensor data. There is a need for 3D data in order to understand dynamic processes such as water infiltration.

L122-L134: Electrical Resistivity Tomography (ERT) method and its applications. ERT monitoring can fill the need for 3D appraisal of soil moisture processes. ERT monitoring technology, such as PRIME, can fill the need for year-round measurements of soil moisture in an Arctic setting.

L135-L142: Science questions.

L143-L155: Description of the manuscript content.

Based on such introductory narrative we formulated two scientific questions (SQs):

SQ1: Considering the need for year-round measurements of Arctic soil properties, and the vulnerability of Arctic soils during winter and shoulder seasons, (a) can geoelectrical sensor technology be used to continuously monitor the coupled heat and water movement (CHWM) in deglaciated sediments year-round? and (b) can we identify and quantify characteristics of CHWM profile in deglaciated sediments during vulnerable periods?

SQ2: (a) Considering the need to understand Arctic pedogenesis post deglaciation, can CHWM differences between sediments at different stages of development since deglaciation be identified? and (b) How do they express in relation to physical properties, location and topography, through processes of freeze-thaw transition and melt water infiltration?

C2: Addressing the lack of scientific structure would help to rectify the problem that the manuscript reads more like a report than a scientific article. My assessment is that the problem has not been corrected – restructuring around a scientific structure that includes a science question and discussion of how that science question has (or not) been answered would help substantially progress away from the 'report' feel that the text currently has.

R2: In order to rectify the lack of scientific structure we have compiled a list of outcomes (O) generated by our work:

1) site-specific Archies Law calibrations
2) soil moisture variations associated with rain-on-snow events
3) site- specific length of the zero-curtain effect
4) site-specific speed, direction, and magnitude of thawing front
5) classification of subsurface regions related to water storage variability
6) older deglaciated sediments experience a longer thaw
7) older deglaciated sediments are dominated by lateral thaw propagation driven by topography and snow
8) younger deglaciated sediments are dominated by vertical thaw

We have now reformulated the text in the conclusions section (L646-669 Track Changes manuscript) in order to reflect our answers to the scientific questions sat out in the introduction. Answers to the two scientific questions are enabled by our results and their interpretation (outcomes labelled when mentioned in text below). The conclusions now read:

"Geoelectrical instrumentation was successful in monitoring the CHWM processes at both sites (i-a). During the Arctic spring, we recorded elevated levels of soil moisture and temperature associated with an anomalous rain on snow event (O2, i-b). During the shoulder period between freeze and thaw, deglaciated sediments experienced the zero-curtain effect. The time the sediments spent around the $0\,°C$ isotherm depends on the snow cover thickness, in this instance determined by the local topography and the site's position on the glacier forefield (O3, i-b). Finally, our ERT monitoring stations were successful in obtaining almost uninterrupted timelapse recordings, which revealed unprecedented 4D images of the Arctic soil freeze-thaw transition. Such records allow one to calculate the speed, direction, and magnitude of the thawing front (O4, i-b). Furthermore, unsupervised k-means clustering proved to be an effective method of classifying regions of the imaged sediment volume according to their electrical

resistivity coefficient of variance, indicating how local site conditions affect water storage variability. Some clusters identified are representative of areas of increased water content whereas others are representative of areas containing higher dense materials, potentially buried ice or rocks (O5, i-b).

Differences in CHWM profiles between sites (ii-a) are underpinned by site specific Archies law calibrations (O1, ii-a) and different thaw front velocities (O4, ii-a). We found that the CHWM profile change between sites is expressed (ii-b) through a different thaw propagation, with older sediments dominated by a longer (O6, ii-b) lateral thaw (O7, ii-b) driven by topography and snow, whereas younger sediments were dominated by vertical thaw (O8, ii-b). Clusters identified at different sites of a similar CV and spatial distribution exhibit a different gradient of average electrical resistivity values, which may again imply that older sediments are subjected to a more time distributed freeze-thaw transition, under conditions closer to thermodynamic equilibrium (O6, ii-b). "

We have further made improvements to the text throughout our discussion sections in order to frame it around the two scientific questions.